# Effective EMI shielding behaviour of thin graphene/PMMA nanolaminates in the THz range

Christos Pavlou [1,2], Maria Giovanna Pastore Carbone [1], Anastasios C. Manikas [2], George Trakakis [1], Can Koral [3], Gianpaolo Papari[4], Antonello Andreone [3,4] & Costas Galiotis [1,2 ✉]

The use of graphene in a form of discontinuous flakes in polymer composites limits the full exploitation of the unique properties of graphene, thus requiring high filler loadings for achieving- for example- satisfactory electrical and mechanical properties. Herein centimetre-scale CVD graphene/polymer nanolaminates have been produced by using an iterative 'lift-off/float-on' process and have been found to outperform, for the same graphene content, state-of-the-art flake-based graphene polymer composites in terms of mechanical reinforcement and electrical properties. Most importantly these thin laminate materials show a high electromagnetic interference (EMI) shielding effectiveness, reaching 60 dB for a small thickness of 33 μm, and an absolute EMI shielding effectiveness close to $3 \cdot 10^5$ dB cm$^2$ g$^{-1}$ which is amongst the highest values for synthetic, non-metallic materials produced to date.

[1] Foundation for Research and Technology Hellas, Institute of Chemical Engineering Sciences, Stadiou St. Platani, Patras, Greece. [2] Department of Chemical Engineering, University of Patras, Patras, Greece. [3] INFN Naples Unit, Naples, Italy. [4] Department of Physics "E. Pancini", University of Naples "Federico II", Naples, Italy. ✉email: c.galiotis@iceht.forth.gr

Since its discovery, graphene has attracted a lot of attention for the development of light-weight, multifunctional composite materials owing to its exceptional mechanical and electronic properties[1–3]. An increasingly growing field of application of such composites is electromagnetic-interference (EMI) shielding, which has become crucial in aerospace, automotive, and portable electronics[4–7]. To date, graphene in the form of nanoparticles, such as nanosheets of graphene oxide (GO), reduced graphene oxide and graphene nanoplatelets (GNPs), has often been employed for the production of polymer composites[8,9]. However, the full exploitation of the properties of graphene is limited in the discontinuous, particle-reinforced composites[10,11], and many issues have to be overcome to make these materials attractive for industrial applications. Typical limitations of discontinuous graphene composites are the small lateral size of graphene flakes that prevents efficient stress transfer from the surrounding matrix[8], the difficulties with particle dispersion and the limited control of flake thickness. Thus high filler loadings are normally required for the attainment of decent electrical and thermal conductivities and for a moderate EMI shielding effectiveness (EMI SE)[12–15].

Nanolaminates, consisting of assemblies of continuous, well-oriented, nano-thin layers, are an emerging class of nanocomposite materials potentially able to exploit the unique multifunctional properties of two-dimensional materials such as graphene. Such structures are designed with different stacking sequences and layer thicknesses, and mimic in effect materials found in nature[16], which exhibit physical properties not often encountered in man-made materials. In light of that, the incorporation of large-size graphene sheets grown by Chemical Vapour Deposition (CVD) in nano-laminated structures of alternating polymer and graphene layers, can be a smart strategy in order to overcome the typical drawbacks of nanoparticle fillers mentioned earlier.

Recently, the inclusion of CVD graphene monolayer into poly (methyl methacrylate) (PMMA)[17] and in polycarbonate (PC)[18] by stacking and folding approaches has been proposed, resulting in a significant increase of elastic modulus and electrical conductivity and giving basic evidence of the potential of continuous graphene nanolaminates. However, the maximum graphene content that could be achieved by stacking method was <0.2%, since the manipulation of centimetre-scale ultra-thin layers may be rather tricky. Also, characterisation of mechanical and electrical properties was presented only for one[17] or very few compositions[18], therefore a full understanding of prevailing trends in these materials is still limited.

Herein we report on the development of centimetre-size CVD graphene-polymer nanolaminates that have the potential to outperform the current overall state-of-the-art graphene-based composite materials in both mechanical and electrical properties per graphene content, and that show high absolute EMI shielding effectiveness in a full frequency decade approximately centred at 1 THz. The fast-growing development of sources, devices and systems presently renders the terahertz region of the electromagnetic spectrum the new frontier in different fields such as data communication, signal modulation, high resolution imaging, and molecule sensing[19–23]. This is leading to a pressing demand for novel and lightweight shielding materials combining small volume and high frequency operation, in order to effectively reduce or eliminate wave interference in tiny and delicate environments. By casting ultra-thin polymer films and combining an iterative 'lift-off/ float-on' process with wet depositions, a large number of polymer/CVD graphene laminae are progressively deposited to construct the nanolaminates at the macro-scale. The process is semi-automatic, scalable and allows for the production of laminates of maximum dimension of 7 cm × 7 cm. We have produced free-standing nanolaminates from CVD monolayer polycrystalline graphene with layer numbers ranging from 10 to 100 and volume fractions of 0.04–0.5%. Moreover, supported nanolaminates with higher graphene content (1 vol%) have also been produced revealing the potential of this class of material. A systematic investigation of the mechanical, electrical and EMI shielding properties of these nanolaminates is presented herein.

## Results and discussion

**Fabrication of CVD graphene/PMMA nanolaminates.** The semi-automatic, iterative 'lift-off/ float-on' process combined with wet depositions proposed herein is schematically depicted in Fig. 1a. We have chosen PMMA as polymeric matrix due to its transparency and good adhesion to graphene[24,25]. By using the proposed all-fluidic manipulation (which is schematically illustrated in the Supplementary Movie 1), CVD graphene/PMMA (Gr/PMMA) layers of thickness much lower than 100 nm can be easily handled and progressively deposited to fabricate nanolaminates with relatively high graphene volume fractions for such thin membranes (Supplementary Table 1 and Supplementary Fig. 1). Furthermore, the process can be scaled-up in both the size of the layers (up to 7 cm × 7 cm) and in the number of simultaneous depositions performed in several transfer devices working in parallel, so that the production of each nanolaminates membrane can be speed-up. Pictures of the $12 \, cm^2$ -nanolaminates with different graphene volume fractions $V_{Gr}$ and SEM images of the cross-section of the sample with 0.13 vol% of graphene are shown in Fig. 1b, c. The SEM image highlights the very regular lamination sequence of the Gr/PMMA laminate, and proves the uniform graphene distribution, with ~250 nm-thick layer of PMMA separated by the CVD graphene sheet. Raman spectroscopy has been adopted for the quality control of graphene during lamination process. As it is clearly presented in Fig. 2, minor spectroscopic changes were observed. In particular, in both the single and multiple Gr/PMMA samples, graphene experiences a small compression (~0.08%) as G peak is slightly blue shifted. Also, the full width at half maximum of the 2D peak, FWHM (2D), is ca. 32 $cm^{-1}$ and the intensity ratio $I(2D)/I(G)$ (Raman 2D-peak to G-peak) is ~2, which are typical values for CVD graphene monolayers[26]. However, a slight increase of the intensity ratio $I(D)/I(G)$ (Raman D-peak to G-peak) in the Gr/PMMA nanolaminates indicates that some structural defects are inadvertently induced by the lamination process.

**Mechanical properties.** The mechanical performance of the produced nanolaminates vis-a-vis the PMMA control has been assessed by means of uniaxial tensile testing and representative stress strain curves are shown in Fig. 3a. It is interesting to note that the produced Gr/PMMA nanolaminates demonstrate a substantial increase of Young's modulus (taken from the initial ~0.4% strain part of the curve) as a function of graphene content and a linear relationship between modulus and volume fraction is found to hold at all graphene loadings for this type of composite (Fig. 3b). The addition of only 0.5% in volume of graphene leads to an increase of stiffness of 250% ca., which is very high compared to typical polymer filled with graphene particles (e.g. GNP, GO, …) for similar or even higher volume fractions[27–32]. This can be ascribed to the efficient reinforcement provided by graphene sheets in the nanolaminate configuration (see Supplementary Discussion and Supplementary Fig. 2). In fact, a literature review about the mechanical behaviour of PMMA reinforced with different graphene particles (Fig. 3b and Supplementary Table 2) clearly reveals that the Gr/PMMA nanolaminates show the highest improvement in terms of Young's

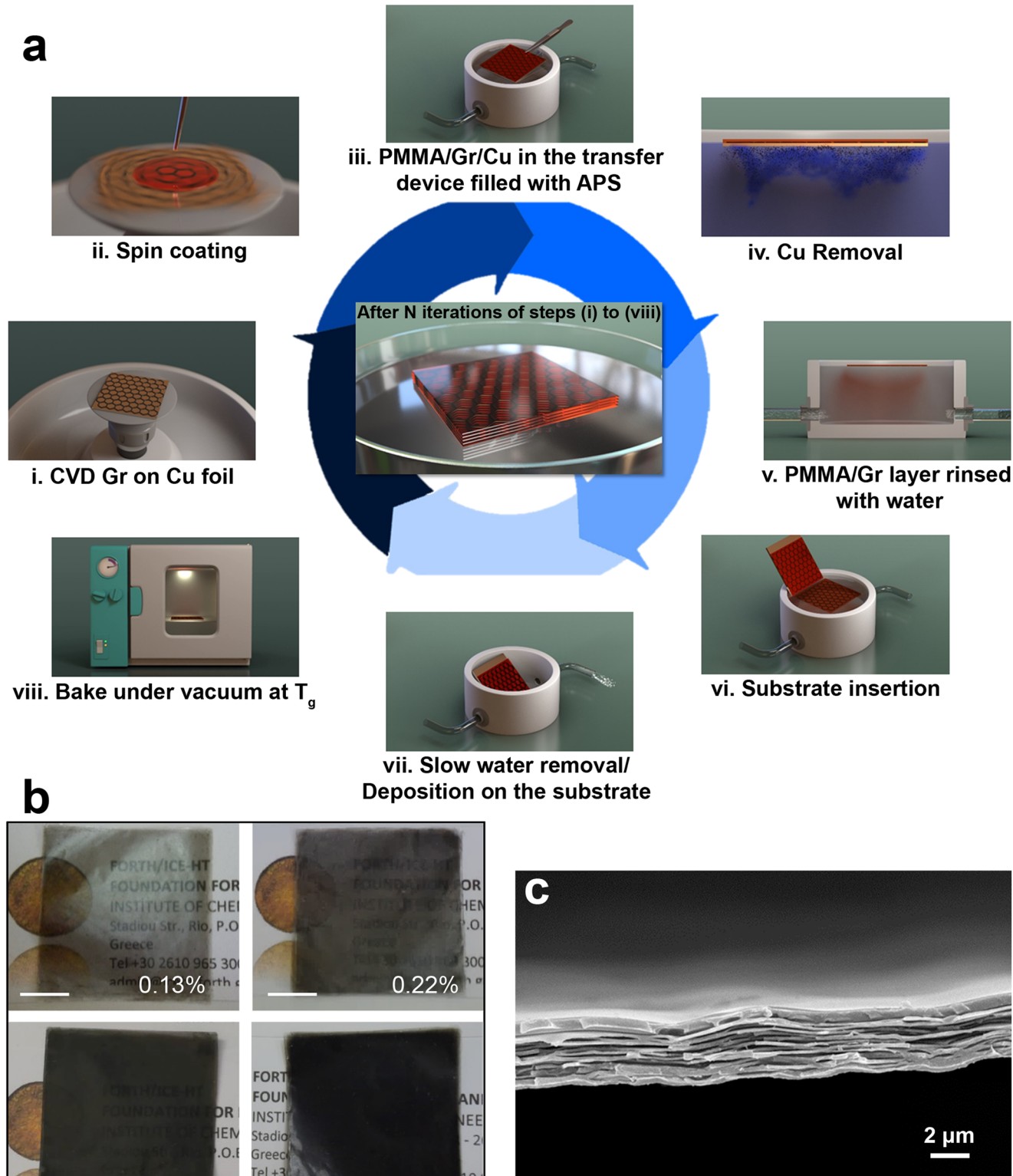

**Fig. 1 Production of CVD graphene/PMMA nanolaminates. a** Schematic illustration of the iterative 'lift-off/ float-on' process combined with wet depositions adopted herein. CVD: chemical vapour deposition. Gr: graphene. PMMA: poly(methyl methacrylate). APS: ammonium persulphate. **b** Representative pictures of the produced centimetre-scale Gr/PMMA nanolaminates with increasing graphene content $V_{Gr}$ (scale bar is 1 cm) and **c** SEM image of the laminate with $V_{Gr} = 0.13\%$ in the cross-section plane, representative of ten experiments.

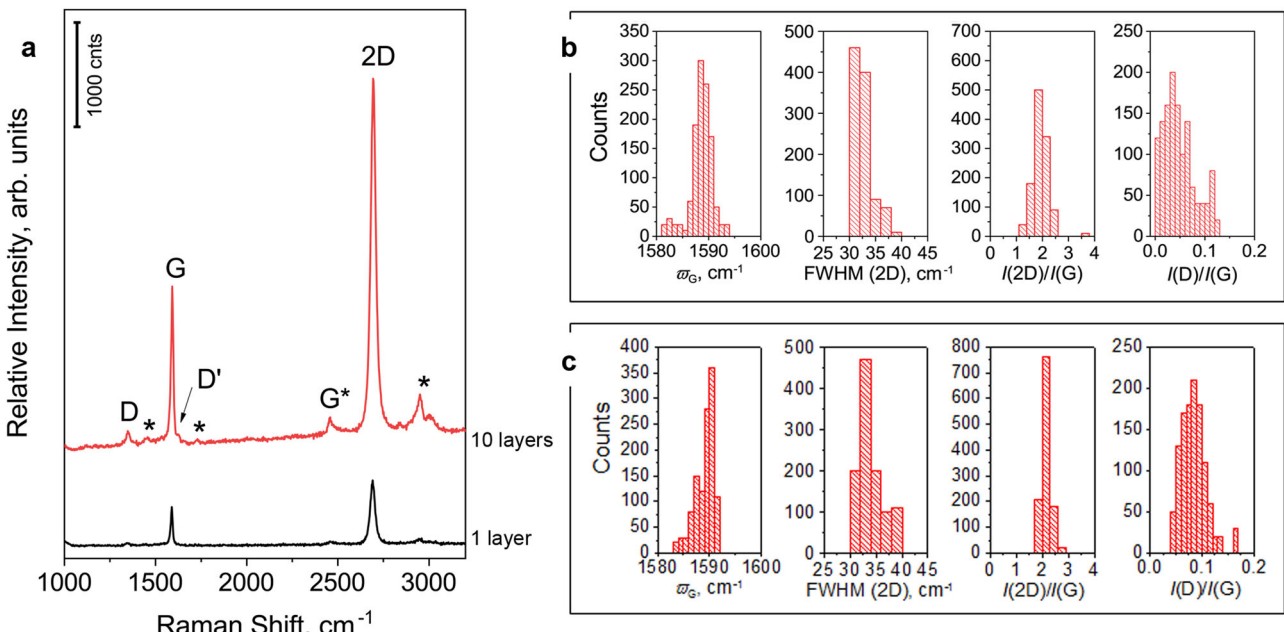

**Fig. 2 Raman spectroscopy investigation of Gr/PMMA nanolaminates. a** Representative Raman spectra collected from single and multiple graphene/PMMA layers showing that monolayer character is retained after multiple depositions. Asterisks mark the spectroscopic features of the poly(methyl methacrylate). **b** Distributions of G peak position ($\omega_G$), full width at half maximum of 2D peak (FWHM(2D)), intensity ratios $I(2D)/I(G)$ and $I(D)/I(G)$ for 1 Gr/PMMA layer and **c** for 10 Gr/PMMA layers.

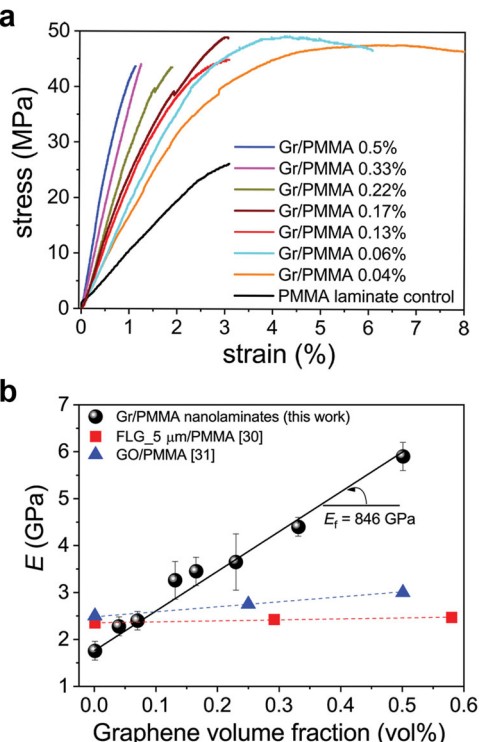

**Fig. 3 Mechanical characterization of Gr/PMMA nanolaminates.**
**a** Representative stress-strain curves obtained by uniaxial tensile testing and **b** Young's modulus (taken from the initial ~0.4% linear part of the curve) of the Gr/PMMA nanolaminates as a function of graphene volume fraction and respective comparison with typical discontinuous graphene composites (data taken from refs. [30,31]). FLG: few-layer graphene. GO: graphene oxide. PMMA: poly(methyl methacrylate). Error bars represent standard deviation.

modulus. Actually, by using the rule of mixture (Supplementary Eq. (1-i)), an effective elastic modulus of 846 GPa has been estimated for graphene in the Gr/PMMA nanolaminates presented herein, which is very close to the Young' modulus estimated for centimetre-scale near single-crystal monolayer graphene[33]. This is significantly higher than effective values for graphene in similar nanolaminates produced by other techniques (e.g. by stacking and folding[18,34], evaluated on the basis of tensile tests performed at even higher strain rates). This proves that our production method leads to regular incorporation of graphene monolayer sheets which are uniformly distributed in the final composite. Moreover, in discontinuous PMMA-graphene composites, the effective Young's modulus of graphene ranges from 15 to 350 GPa[27–32], depending on particle type, lateral size, composite production process and functionalization. This is not surprising since, as mentioned earlier, in discontinuous composites the maximum reinforcement is limited by the small lateral size of the filler and aggregation/orientation issues[11]. Also, the Gr/PMMA nanolaminates present a significant improvement of strength (around 100%) compared to neat PMMA laminate. Nevertheless, it is worth adding that, for graphene content over 0.22 vol%, an embrittlement of the nanolaminates is observed which could be ascribed to defects originated at random locations in the non-uniform thickness of the ultra-thin polymer/graphene layers[35]. Actually, optical microscopy and AFM images (Supplementary Fig. 3) reveal that the single polymer/graphene layer conformally reproduces the morphology of the Cu foil after CVD graphene growth, which presents typical roughness values around 70 nm or even more[36,37]. As the thickness of the polymer layer decreases (e.g. to 65 nm for $V_{Gr} = 0.5$ vol%), the contribution of such morphological defects contributes to the premature failure of the entire nanolaminate under tensile loading. Furthermore, the increased number of depositions in the lamination procedure (up to 100 in the laminate with 0.5 vol% graphene) may also introduce ruptures and tears in graphene that can affect its strength, in agreement with the increase of the

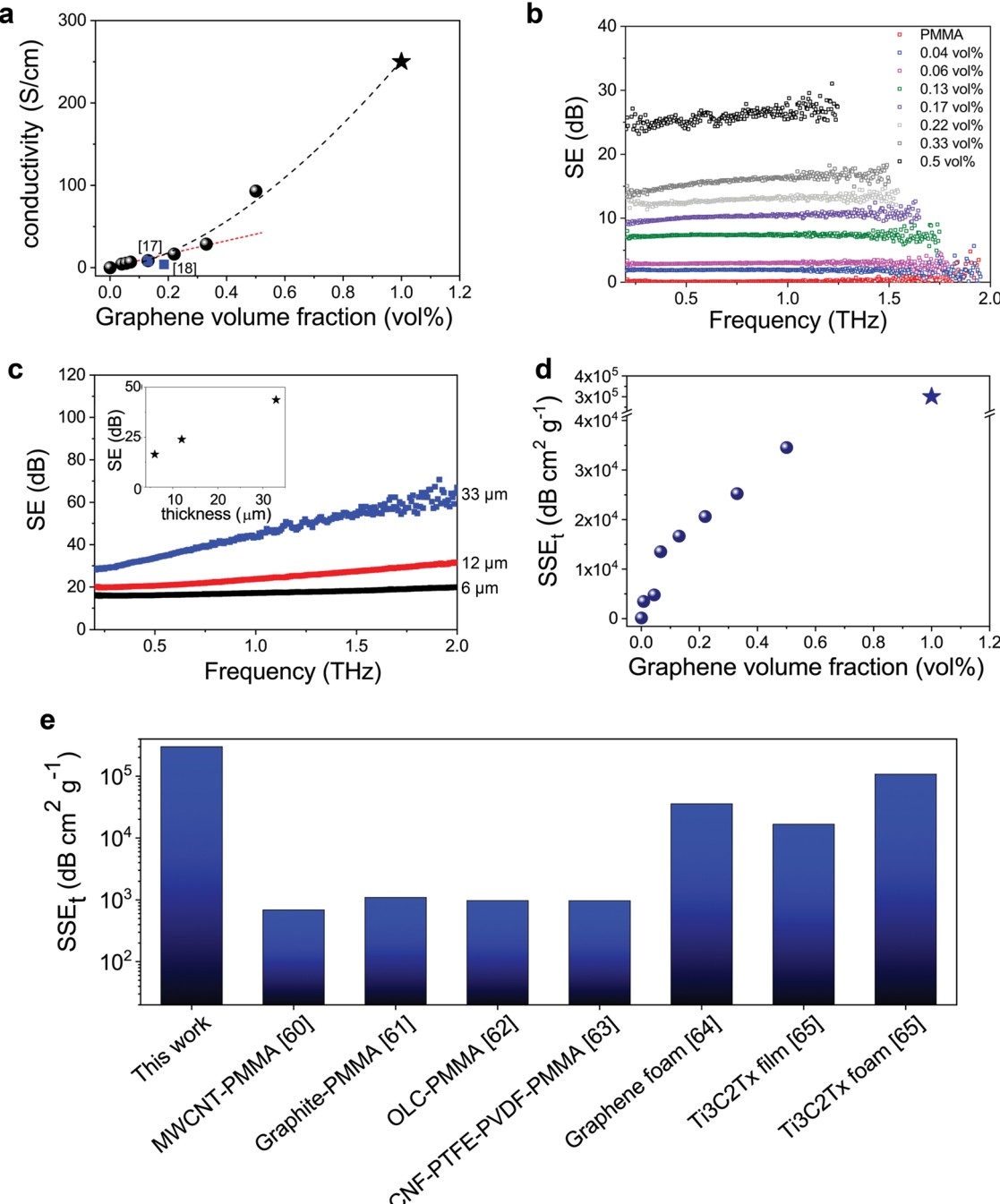

**Fig. 4 Electrical and EMI shielding properties of the Gr/PMMA nanolaminates. a** In-plane electrical conductivity of the nanolaminates as a function of graphene volume fraction. Red short dashed line represents the linear fitting of the experimental data for low graphene content and black dashed line represents the fitting of the experimental data to $y = ax^b$, with $a = 251$ S/cm and $b = 1.63$. Blue square symbols represent reference data for similar systems [17,18]. Star represents the conductivity value measured on a model 1 vol% nanolaminate supported on a quartz substrate. **b** Shielding effectiveness of the freestanding nanolaminates as a function of frequency. SE: shielding effectiveness. **c** Shielding effectiveness of nanolaminate with 0.33 vol% of graphene in the same experimental frequency range as in (**b**), for specimens with different total thicknesses. Inset: SE measured at 1 THZ as a function of thickness. **d** Absolute shielding effectiveness of the nanolaminates as a function of graphene volume fraction. SSE$_t$: absolute shielding effectiveness. Star symbol represents the SSE$_t$ value measured on a model 1 vol% nanolaminate supported on a quartz substrate. **e** Comparison of SSE$_t$ values between Gr/PMMA nanolaminates and state-of-the-art shielding materials in the THz range [60–65]. PMMA: poly(methyl methacrylate). MWCNT: multi-walled carbon nanotubes. OLC: onion-like carbon. CNF: carbon nanofiber. PTFE: polytetrafluoroethylene. PVDF: poly(vinylidene fluoride). Ti3C2Tx: Titanium Carbide.

Raman D/G intensity ratio mentioned earlier. Production of nanolaminates by using higher quality graphene grown on Cu foil with lower roughness is in progress and is expected to improve this behaviour, as shown by preliminary tests reported in Supplementary Fig. 4.

**Electrical properties.** Graphene electrical conductivity has been preserved in the Gr/PMMA nanolaminates during the lamination process, as demonstrated by in-plane electrical measurements (Fig. 4a). In fact, the produced nanolaminates possess anisotropic conduction, with in-plane conductivity ranging from 8.7 S/cm

(for 0.13 vol%, in agreement with value reported in ref. [17]) to 95 S/cm (for 0.5 vol%) which, to our knowledge, is one of the highest values reported for polymers filled with graphene or graphene-related materials[38]. Actually, electrical conductivity of PMMA filled with graphene particles can be several order of magnitudes lower than those measured for nanolaminates with the same graphene content (see Supplementary Table 2), due to issues in obtaining uniform dispersion of the graphene filler inside the polymer matrix without aggregation[27,38,39]. The very high electrical conductivity obtained in the Gr/PMMA nanolaminates can be attributed to both the high quality of the graphene monolayer produced via CVD and to the laminated structure, where continuous, large graphene sheets are well dispersed within the matrix. In fact, unlike particular-filled composites in which electrical conductivity is limited by electron hopping[38,40–42], the electrical conduction in the proposed nanolaminates is due to fast electron transport in the π bonds of the graphene sheets that are separated by the continuous polymer layers. Hence, in this laminated architecture of perfectly oriented graphene layers, each of them should retain its intrinsic properties, and the material can be modelled as a two-dimensional parallel system of conducting sheets with an equivalent sheet conductance $N$ times that of an isolated graphene layer[43]. As a proof of that, the contribution of graphene to the electrical conductivity has been evaluated from linear fitting to be $8.1 \times 10^3$ S/cm, which is not far from reported values for CVD monolayer graphene transferred on Si wafer[44,45]. It is worth noting that the effective conductivity of graphene in our nanolaminates is higher than in earlier attempts[17], proving the better quality of the structure obtained by the process proposed herein. However, we observe that the linearity of in-plane conductivity with volume fraction is lost for higher graphene content (i.e. for $t_{PMMA} < 100$ nm), which may suggest the occurrence of further mechanisms contributing to charge transport. Actually, the aforementioned inhomogeneity of the polymeric layer thickness due to the use of a rough sacrificial substrate probably causes interconnection or bridging between adjacent layers. At this stage, therefore, it is possible that the local thickness of the PMMA spacer is comparable with the cut-off distance for electron jumping between two parallel graphene sheets insulated by polymers ($<5$ nm[46–48]) and hopping electrons are triggered thus enhancing the micro-current in the layered package of graphene. Under these conditions, the conduction mechanism could be ascribed to the combination of migrating and hopping electrons[38,40,41,49]. It is important to underline that the current maximum for free-standing graphene-polymer nanolaminates produced in this work is achieved at 0.5 vol%; however, there is still room for improvement for higher graphene volume fractions by further reducing the thickness of PMMA interlayer. In fact, several studies demonstrated that stable ultra-thin PMMA films with thicknesses of 20–30 nm can be produced by spin coating[50]; therefore, in principle, the proposed iterative 'lift-off/ float-on' process enables the fabrication of laminates with even higher content of graphene and hence higher values of electrical conductivity. In order to prove that, model nanolaminate specimens with 1 vol% of graphene were fabricated by depositing a limited number of Gr/PMMA layers on quartz substrate (i.e. 4 layers, with polymeric interlayer of 33 nm ca.) and an average in-plane electrical conductivity of 250 S/cm has been measured (Fig. 4a).

**EMI shielding in the THz range.** Materials with large electrical conductivity are typically required to obtain high EMI shielding performance. Therefore, in light of the high electrical conductivity of the Gr/PMMA nanolaminates, and inspired by the pioneering theoretical works on multilayer graphene screens[51–53] and previous studies on graphene-like materials[54], we assessed the shielding properties in the range 0.2–2 THz, across one frequency decade[55]. Terahertz shielding is becoming increasingly important since there has been recently a significant advancement in the development of very high frequency electronics and devices for different applications, such as wireless communication, imaging, and sensing. In particular, there is a need for small-volume, light, and very efficient screening systems to protect a specific component and its surroundings.

As already observed for mechanical and electrical properties, the nanolaminate configuration provides better shielding performance compared to discontinuous composites, such as PMMA filled with similar content of graphene-related materials. THz transmission spectra for several freestanding Gr/PMMA nanolaminates with an average thickness of ca. 5 μm are presented in Supplementary Fig. 5, whereas results in terms of EMI shielding efficiency (SE) vs frequency are shown in Fig. 4b. Adding graphene content, the SE values of the Gr/PMMA nanolaminates increase up to 25 dB at 1 THz (for 0.5 vol%). The observed increase of SE compares well with what is theoretically expected in the limit of zero frequency (see related Supplementary Discussion). Furthermore, we show for comparison the same measurements performed on neat PMMA and PMMA filled with 0.27 vol% GNPs in Supplementary Fig. 6. Both the neat PMMA and PMMA/GNP composite are found to be nearly transparent to the THz waves, with SE values always lower than 0.2 dB in the experimental frequency range.

Indeed, the EMI shielding of the polymer-CVD graphene nanolaminates originates from the intrinsic high electrical conductivity of graphene being conserved under stacked geometries[43]. This results in a dynamic and complex relationship amongst different terms composing the total shielding effectiveness, namely absorption, (surface) reflection, and internal multiple reflections, strongly dependent on the nanolaminate geometrical parameters such as the number of layers and interlayer distance[56] (see Supplementary Fig. 7 and Supplementary Discussion). For all samples having same average thickness and different graphene content, SE is found to be only slightly frequency dependent in the investigated THz range, since the contribution to SE given by the reflection and absorption terms are comparable, with an opposite trend as a function of frequency (see Supplementary Eqs. (5) and (6)). Furthermore, for a fixed graphene volume fraction (0.33 vol%), we measured the effect on the total shielding produced by augmenting the contribution given by absorption, by changing the overall thickness of the nanolaminate membrane from 6 μm up to 33 μm (Fig. 4c). Since losses here are the dominant shielding mechanism, SE as a consequence increases with frequency, reaching a value of 60 dB at 2 THz for the thickest sample. Note also in the inset that at a given frequency (1 THz in the plot) a linear increase with thickness is observed, as expected from Eq. (S6).

Obviously, since SE depends on the material thickness, adequate shielding can be achieved with thicker layers but at the expense of adding extra weight to the system. This is normally unacceptable for moving parts as those encountered in aerospace applications. Furthermore, with the development of compact electronic devices, the requirements for EMI shielding materials are moving toward light-weight, flexible systems capable to exhibit strong absorption per unit volume and/or weight. It is clear that the current challenge in EMI shielding for aerospace and electronics is to achieve high EMI SE with low added weight and at small thicknesses.

In order to assess the EMI shielding performance of different materials per unit weight and thickness, we have divided the EMI SE by the specimen density and thickness, obtaining in such a way the absolute shielding effectiveness of the material ($SSE_t$, measured in dB cm$^2$ g$^{-1}$)[57–59]. The $SSE_t$ values at 1 THz for the

freestanding Gr/PMMA nanolaminates are plotted in Fig. 4d as a function of graphene volume fraction. In the same plot, we added the results of shielding measurements performed on the model nanolaminate with 1 vol% deposited on a quartz substrate (nearly transparent in the THz region). From the plot it is evident that our material shows very high $SSE_t$, with the potential to achieve values close to $3 \times 10^5$ dB cm$^2$ g$^{-1}$. As shown in Fig. 4e, this value is higher than $SSE_t$ reported for polymer filled with carbon-based fillers[60–63], graphene foams[64] and other non-metallic shielding materials[4]. Therefore, with only a tiny amount of large-size graphene, the produced Gr/PMMA nanolaminates can be placed among the most efficient EMI shielding material systems known to date, including the recently developed Ti$_3$C$_2$T$_x$ films and foams[65]. This finding is notable since in our sample several commercial requirements for an EMI shielding product are engrained in a single material, such as high EMI SE, low density, small thickness and mechanical integrity. Thanks to this unique combination, the nanolaminates presented herein clearly surpass the performance of the light-weight graphene-based foams as best candidates in EMI shielding, since the latter suffer from difficulties in achieving small thicknesses and from the mechanical fragility due to the large and disordered pore structures. Furthermore, unlike state-of-the-art materials which have reached the limit of their potential (e.g. bulk metals, graphene foams[64] or MXenes systems[65]), the shielding performance of the CVD graphene nanolaminates presented herein still has room for improvement by either varying graphene content and/or increasing the number of layers.

In summary, we have exploited a bottom-up approach based on wet transfer and subsequent depositions to produce light-weight, stiff and electrically-conductive large-size CVD graphene/polymer nanolaminates, with very high values of absolute EMI shielding effectiveness. Ultra-thin film technology has been adopted here to decrease the thickness of the polymer layer up to less than 100 nm in order to achieve graphene content higher than proposed earlier[17,18,34]. The produced free-standing nanolaminates present an enhancement of Young's modulus, with an estimated effective modulus of 846 GPa for CVD graphene, combined with in-plane electrical conductivity as high as 95 S/cm. As evidenced by measurements on model supported specimens, an important aspect of this work is that by further increasing the volume fraction, graphene nanolaminates have the potential of exhibiting unmatched values across the whole spectrum of physical-mechanical properties. Actually, our results point to in-plane electrical conductivity and absolute shielding effectiveness values amongst the highest ever-reported in literature and over a frequency decade in the THz region. These findings pave the way to the development of graphene polymer nanolaminates for a whole range of applications in the high frequency regime, including data communication, electronics and aerospace.

## Methods

**Nanolaminates production**. Graphene was grown on 7 cm × 7 cm Cu sheets (JX Nippon Mining & Metals, 35 μm-thick, 99.95%) in a commercially available CVD reactor (AIXTRON Black Magic Pro, Germany). Then, continuous graphene laminates were prepared by semi-automatic, sequential transfer of Gr/PMMA layers onto the top of the previous layer using an iterative 'lift-off/ float-on' process combined with wet depositions. In the process, we exploit the Cu foil adopted for graphene growth also as sacrificial layer and specific transfer devices were designed in order to control the flow of the fluids during each deposition. The transfer system consisted of conical polytetrafluoroethylene (PTFE) tanks equipped with control flow devices to regulate the fluid inlet and outlet. The production of each PMMA/Gr layer was done by spin coating a PMMA solution in anisole (495 PMMA, Microchem) on the graphene grown on copper foil of dimensions 35 mm × 35 mm. This step is fundamental for the control of the graphene volume fraction ($V_{Gr}$) in the nanolaminate. In fact, $V_{Gr}$ is defined as $\frac{t_{Gr}}{t_{Gr}+t_{PMMA}}$, being $t_{Gr}$ the thickness of monolayer graphene (0.334 nm[66]) and $t_{PMMA}$ the thickness of the PMMA layer. However, since $t_{Gr}$ is always much smaller than $t_{PMMA}$, then the graphene volume fraction expresses also the thickness ratio between graphene and PMMA for each specimen.The spinning conditions and concentration of the solution were preliminarily optimised to fabricate films with specific thickness, compatible with the desired graphene content in the nanolaminate (see Supplementary Table 1 and Supplementary Fig. 1). In this optimisation process, the thickness of the single PMMA/graphene layer deposited on a Si wafer was measured by using Atomic Force Microscopy (see Supplementary Methods) according to the scratch step method[67]. After spin coating, the copper was etched away using a 0.1 M aqueous solution of ammonium persulphate (APS). Afterwards, the floating PMMA/Gr layer was rinsed with deionized-double distilled water inserted in the transfer device until full replacement of the APS solution. By slowly reducing the water level, the floating PMMA/Gr layer was deposited on another PMMA/Gr layer on a copper foil, which represents the substrate for subsequent depositions. After each deposition, the multi-layer was firstly dried at 40 °C for some hours under vacuum to remove the excess of water and then, in order to improve adhesion between subsequent layers, a post-bake process was performed at 150 °C for 5 min on a hot plate. In order to scale up and accelerate the production of nanolaminates, the etching/ lift-off steps were simultaneously performed in several transfer devices. The cycle was repeated until the desired number of PMMA/Gr layers was reached (Supplementary Table 1). At the end, the Cu substrate was etched away in APS solution to release the freestanding PMMA/Gr nanolaminate, then rinsed with water and finally dried at 40 °C under vacuum to remove the excess of water. By using a similar procedure, a control nanolaminate of neat PMMA was also produced. Model specimens with 1 vol% of graphene were fabricated by depositing 4 Gr/PMMA layers (each one of thickness ~33 nm) on a quartz substrate.

**Uniaxial tensile testing**. Tensile test was performed using a micro-tensile tester (MT-200, Deben UK Ltd, Woolpit, UK) equipped with a 5 N load cell. Prior to the test, samples were cut into stripes having an overall length of 35 mm, a gauge length of 25 mm and a width of 1 mm. For each specimen, the thickness was determined as the mean of 10 measurements along the gauge length with a digital micro-metre with a resolution of 0.1 μm (Mitutoyo, Japan). All test specimens were secured onto paper testing cards using a two-part cold curing epoxy resin (Araldite 2011, Huntsman Advanced Materials, UK) to avoid damages to the gripping area. The specimens were subsequently loaded in tension with a crosshead displacement speed of 0.2 mm min$^{-1}$ (corresponding to a test specimen strain rate of 0.008 min$^{-1}$). Stress and strain were calculated based on the measured machine-recorded forces and displacements. The Young's modulus was estimated through a linear regression analysis of the initial linear portion of the stress–strain curves (~0.4% strain). Average results of 10 test specimens are reported for each sample.

**Raman spectroscopy**. Raman mapping has been performed on an area of 100 × 100 μm$^2$, by acquiring spectra at steps of 3 μm. A Renishaw Invia Raman Spectrometer with 2400 and 1200 grooves/mm grating for the 514 nm laser excitation and a 100× lens were used for the evaluation of graphene quality on macroscale nanolaminates.

**Electrical conductivity measurements**. The electrical resistance was measured in a four-point scheme on stripes cut from the nanolaminates, and the conductivity was calculated from specimen dimensions.

**EMI shielding in the THz range**. The EMI shielding response of the nanolaminates was investigated in the range 0.2–2 THz by using a time domain spectrometer (TeraK15 from Menlo Systems) driven by a femtosecond laser fibre-coupled to photoconductive antennas both for THz emission and detection. Data acquisition was realised by means of a lock-in amplifier coupled with electronics and computer software. A standard setup with four polymethypenetene lenses was used to collimate and focus the beam first impinging onto the sample plane and then received by the detector. Computer aided motion controller was utilised for accurate target positioning. All samples have been tested under dry conditions (<0.1% relative humidity) in a nitrogen environment to avoid spurious effects caused by moisture. Samples are placed on an Al sample holder having a hole of radius 8 mm. Since the THz beam waist $w_0$ at the focus point is approximately 1.5 mm, the area under illumination is at most only 10% of the sample effective surface, making negligible any scattering given by edge or wedge effects. Since $(\mathbf{k}_0 w_0)^2$ is much larger than unity in the investigated frequency range ($\mathbf{k}_0$ being the wavevector in free space), we can safely assume that the plane wave approximation holds for all measurements that have been carried out within a few percent relative error[68]. The THz beam impinges on the sample surface under normal incidence. The sample is placed on a kinematic mount for fine adjustment and the sample surface parallelism with respect to the beam wavefront is controlled by a three-points calibration method. Therefore, paraxial error is considered to be very small. The electric field versus time was acquired separately upon transmitting through the sample ($\hat{\mathbf{E}}_{smpl}$) and through the reference material ($\hat{\mathbf{E}}_{ref}$, air in case of freestanding samples, a 1 mm-thick quartz substrate in case of the 1 vol% nanolaminate). In all cases, the reference measurements were performed under the same experimental conditions applied to the samples, namely with the pulsed signal passing through the holed Al

holder. Time domain signals, averaged over 3000 waveforms, are recorded over a relatively large time temporal interval up to 300 ps and then converted into the frequency domain by applying a Fast Fourier Transform (FFT). From here the electric-field ratio, the transmittance $T(\omega) = \left| \frac{\tilde{E}_{smpl}(\omega)}{\tilde{E}_{ref}(\omega)} \right|$, was measured with a resolution better than 5 GHz.

## Data availability

The authors declare that the data supporting the findings of this study are available within the article and its supplementary information files. All other relevant data are available from the corresponding author upon reasonable request.

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

## Acknowledgements

The authors acknowledge the financial support of the research project Graphene Core 3, GA: 881603 and National Institute for Nuclear Physics (INFN) under the project "TERA". The authors thank Graphenea S.A. and, in particular, Dr. Amaia Zurutuza and Dr. Alba Centeno for kindly supplying CVD graphene at the early stage of the project. The authors also are grateful to Dr. Georgia Tsoukleri and Mr. George Paterakis for useful discussion and technical support.

## Author contributions

C.P., M.G.P.C., A.C.M. and G.T. prepared and characterised the samples, in terms of Raman, mechanical and electrical experiments. C.K., G.P. and A.A. performed the EMI shielding experiments and analysed the data. Finally, C.G., M.G.P.C., A.C.M. and A.A. analysed the collected data and wrote the manuscript. C.G. conceived the idea and supervised the entire project.

## Competing interests

The authors declare no competing interests.
