## [Peer Review File · Nature Communications]

REVIEWER COMMENTS

Reviewer #1 (Remarks to the Author):

This paper describes a method to produce graphene-polymer laminates with enhanced mechanical, electrical and electromagnetic properties, for specific applications in EMI shielding in the THz range. The paper major claims are:

- 1) Production centimetre-size CVD graphene-polymer nanolaminates that have the potential to outperform the current overall state-of-the-art graphene-based composite materials in both mechanical and electrical properties per graphene content;
- 2) Graphene content in the nanolaminate higher than 0.2% vol and up to 1% vol;
- 3) The nanolaminate shows impressively high absolute EMI shielding effectiveness in a full frequency decade approximately centred at 1 THz;
- 4) The process is semi-automatic, scalable and allows for the production of laminates of maximum dimension 7 cm × 7 cm.

The novelty of the claim is limited to the technological process which enable the production of centimetre size nanolaminates.

Instead, the idea of realizing graphene-polymer laminate as high-shielding materials is not a new contribution of this paper, but it was originally proposed in the pioneering works of D'Aloia et al., which should be cited in the reference list:

- D'Aloia, A.G., D'Amore, M., Sarto, M.S., "Optimal terahertz shielding performances of flexible multilayer screens based on chemically doped graphene on polymer substrate", (2015) IEEE International Symposium on Electromagnetic Compatibility, 2015-September, art. no. 7256309, pp. 1030-1035.
- D'Aloia, A.G., D'Amore, M., Sarto, M.S., "Terahertz Shielding Effectiveness of Graphene-Based multilayer Screens Controlled by Electric Field Bias in a Reverberating Environment", (2015) IEEE Transactions on Terahertz Science and Technology, 5 (4), art. no. 7128760, pp. 628-636.
- D'Aloia, A.G., D'Amore, M., Sarto, M.S., "Low-Terahertz Modeling of Graphene/Dielectric Multilayers Using an Equivalent Single Layer in Reverberation Environment", (2018) IEEE Transactions on Electromagnetic Compatibility, 60 (4), art. no. 8114264, pp. 849-857.
- D'Aloia, A.G., D'Amore, M., Sarto, M.S., "Tunable graphene/dielectric laminate for adaptive low-gigahertz shielding and absorbing screens", (2018) IEEE Electromagnetic Compatibility Magazine, 7 (2), pp. 82-87.

In fact the cited works 15 and 16 in the reference list concerns the development of graphene-polymer laminates having dimensions up to 2 cm x 2 cm. These works investigate the mechanical and electrical conduction properties of these laminates, but they do not investigate the EMI shielding properties.

Therefore, the novel contribution of this paper is related to the experimental demonstration of the feasibility and performance the original graphene-polymer shielding laminate structure described in the aforementioned papers.

The results may be of interest for application in the field of metal-free multifunctional materials for electromagnetic shielding.

There are some aspects that should be further addressed before accepting the paper for publication.

- 1) The measurement set up for the assessment of the SE at THz should be better described. It should be specified which is the EM field configuration considered for the measurement of the SE (i.e. plane wave? Normal incidence?), which the calibration configuration considered, how the effect of edge and wedge scattering is considered or corrected.
- 2) It is demonstrated that the produced Gr/PMMA nanolaminates demonstrate a substantial increase in Young's modulus as a function of graphene content and, for the first time, it is shown that there is a linear relationship between modulus and volume fraction. A prediction model should be used to validate this result.
- 3) The performance of the shield is described using the parameter SSE, which however does not have a physical or technical limit. In the electromagnetic compatibility community, a parameter which is considered in order to predict the minimum shielding performance of a shield is the limit for frequency attending to zero of the SE, which is directly expressed in terms of the sheet resistance of the screen. It is highly recommended that the authors use this model.

The paper may influence thinking in the field, since it present a model shielding material. However,

the part related to the design of the panel is completely missing. For this reason, a comparison between modelling and experimental testing should be included in order to validate the obtained results.

Reviewer #2 (Remarks to the Author):

In this work, the authors proposed a kind of method to fabricate centimeter-scale CVD graphene (Gr)/PMMA nanolaminates, and investigated the effect of different volume fractions of graphene on mechanical, electrical and electromagnetic properties. The maximal EMI SE reaches 60 dB at a thickness of 33 μm . This work is meaningful. Therefore, I recommend it to be published in Nature Communications after some revision.

(1) The authors fabricated Gr/PMMA layers with different volume fractions of graphene and demonstrated the great effect of volume fraction on mechanical, electrical and electromagnetic properties, but there are in the lack of detailed description to fabricate Gr/PMMA layers with different volume fractions and to measure the volume fractions. This makes me confused.

(2) The crystal structure of single Gr/PMMA is different from that of 10 Gr/PMMA. And the authors said that the ruptures and tears will occur in graphene when the number of depositions increases. Why? What causes the phenomena?

(3) Authors said that the thin polymer layers act as spacers whereas the graphene monolayers tend to conserve the intrinsic properties of isolated graphene, additively contributing to the overall in-plane electrical conductivity of the laminates. What is the meaning of "additively contributing to the overall in-plane electrical conductivity of the laminates"? Please explain it in detail, and analyze the reason.

(4) This is a special alternant nanolaminate structure. I suggest the authors could provide the SER and SEA to further analyze the response mechanism.

(5) The background of the manuscript should be further highlighted over the reported literatures. Some important and relevant literatures should not be ignored, such as Nat. Commun., 2015, 6, 6628; Adv. Mater. 2020, 2002112; Carbon 2009, 47, 922-925; Carbon, 2019, 155, 232-242.

(6) Some experiments and literature have proved that graphene/PMMA nanolaminates have better mechanical properties, but this manuscript lacks theoretical analysis or mechanism elaboration on mechanical properties enhancement.

(7) In the manuscript, the authors said "..... by using the rule of mixture, an effective elastic modulus of 846 GPa.....". The calculation process should be added, which can be put in Supporting Materials.

(8) Compared with graphene/polymer composites, the conductivity of the graphene/PMMA nanolaminates is raised by several orders of magnitude. But in the manuscript, the authors just briefly attributed the enhancement to graphene's distribution state in the polymer, without involving the essence of electron transport. This is not sufficient to raise wide readers' interest. Please provide a further analysis. Actually, there are several important models that can be used to explain. For example, according to the two important models of "Electron-Hopping (EHP)" and "Aggregative-induced-charge transport (AICT)" [Carbon 2010, 48, 788-796; Carbon 2013, 65, 124-139; Adv. Mater. 2014, 26, 3484-3489.], this phenomenon can be well explained that the conductivity of dispersed graphene system is limited by electron hopping, while that of graphene/PMMA nanolaminates is promoted by fast electron migrating due to the continuous π bond.

(9) In this work, the absolute shielding effectiveness (SSET) is used to compare the EMI SE of graphene/PMMA at different thicknesses. However, SET is also relevant to thickness ($e\text{-}ad$), so that the usage of the SSET seems to be not appropriate as the assessment standard.

(10) In this work, authors found that the linearity of in-plane conductivity with volume fraction is lost for higher graphene content, and attribute it to the occurrence of further mechanisms, contributing to electrical conduction. This new mechanism may be the creation of interconnection or bridging between adjacent layers. How does the creation of interconnection or bridging between adjacent layers cause such changes in conductance? Can authors give a clear explanation and

provide some relevant literatures?

(11) According to the equation S1 and S2, the SE increase with frequency. What physical mechanism inside the graphene/PMMA nano-laminates causes this change?

Reviewer #3 (Remarks to the Author):

Recommendation: Reject

This paper by Pavlou et al reports the synthesis/assembly of graphene/PMA nanolaminates via a 'lift-off/ float-on' process for THz EMI shielding. The nanolaminated structure reached EMI SE of 60 dB for a small thickness of 33- μm . Although the EMI SE results are impressive, the methodology of nanolaminated structure is not new and there is no significant attraction in the results which can merit publication in Nature Communications. I will suggest this article for publication in Scientific Reports, after addressing following comments.

1. The authors have published the EMI shielding results of same work earlier in IEEE paper <https://ieeexplore.ieee.org/stamp/stamp.jsp?arnumber=8874200> in 2019. Since the shielding values are the main object of these works, the current submission provides no new insight on shielding values.
2. Although the nanolaminated structure provides good shielding, the process is quite complex and slow as compared to other composite synthesis process, for example, development of foam structures with very low graphene filler content in polymer matrices.
3. The results largely depend on the control of polymer thickness layer via spin coating. How this process can be scalable to meet commercial requirements?
4. For higher graphene content linearity in electrical conductivity is lost. The authors shall state the proper reason, why in-plane electrical conductivity, in the presence of a polymer spacer tends to increase significantly at higher graphene layers? The electrical conductivity mechanisms may be explained.
5. The process of increasing the overall thickness (Figure 4c) is not explained in experimental section.
6. The SEM image in Figure 1c does not reveal any information related to graphene. The layers shown are probably due to polymer spacers; how many graphene layers are expected in, for example 0.3 vol % nanolaminated structure?

Reply to Reviewers' Comments

Reviewer #1

This paper describes a method to produce graphene-polymer laminates with enhanced mechanical, electrical and electromagnetic properties, for specific applications in EMI shielding in the THz range. The paper major claims are:

- 1) Production centimetre-size CVD graphene-polymer nanolaminates that have the potential to outperform the current overall state-of-the-art graphene-based composite materials in both mechanical and electrical properties per graphene content;
- 2) Graphene content in the nanolaminate higher than 0.2% vol and up to 1% vol;
- 3) The nanolaminate shows impressively high absolute EMI shielding effectiveness in a full frequency decade approximately centred at 1 THz;
- 4) The process is semi-automatic, scalable and allows for the production of laminates of maximum dimension 7 cm × 7 cm.

The novelty of the claim is limited to the technological process which enable the production of centimetre size nanolaminates. Instead, the idea of realizing graphene-polymer laminate as high-shielding materials is not a new contribution of this paper, but it was originally proposed in the pioneering works of D'Aloia et al., which should be cited in the reference list:

- D'Aloia, A.G., D'Amore, M., Sarto, M.S., "Optimal terahertz shielding performances of flexible multilayer screens based on chemically doped graphene on polymer substrate", (2015) IEEE International Symposium on Electromagnetic Compatibility, 2015-September, art. no. 7256309, pp. 1030-1035.
- D'Aloia, A.G., D'Amore, M., Sarto, M.S., "Terahertz Shielding Effectiveness of Graphene-Based multilayer Screens Controlled by Electric Field Bias in a Reverberating Environment", (2015) IEEE Transactions on Terahertz Science and Technology, 5 (4), art. no. 7128760, pp. 628-636.
- D'Aloia, A.G., D'Amore, M., Sarto, M.S., "Low-Terahertz Modeling of Graphene/Dielectric Multilayers Using an Equivalent Single Layer in Reverberation Environment", (2018) IEEE Transactions on Electromagnetic Compatibility, 60 (4), art. no. 8114264, pp. 849-857.
- D'Aloia, A.G., D'Amore, M., Sarto, M.S., "Tunable graphene/dielectric laminate for adaptive low-gigahertz shielding and absorbing screens", (2018) IEEE Electromagnetic Compatibility Magazine, 7 (2), pp. 82-87.

In fact the cited works 15 and 16 in the reference list concerns the development of graphene-polymer laminates having dimensions up to 2 cm x 2 cm. These works investigate the mechanical and electrical conduction properties of these laminates, but they do not investigate the EMI shielding properties. Therefore, the novel contribution of this paper is

related to the experimental demonstration of the feasibility and performance the original graphene-polymer shielding laminate structure described in the aforementioned papers. The results may be of interest for application in the field of metal-free multifunctional materials for electromagnetic shielding. There are some aspects that should be further addressed before accepting the paper for publication.

Authors' reply

We thank the Reviewer for his/her comment. In light of that, we have now highlighted in the text that the study of the terahertz shielding of the graphene-PMMA nanolaminates has been inspired by the theoretical studies on multilayer graphene screens by D'Aloia et al. and we have updated the reference list. The amended manuscript, on page 8 lines 10-12, now reads:

“Therefore, in light of the remarkable electrical conductivity of the Gr/PMMA nanolaminates, and inspired by the pioneering theoretical works on multilayer graphene screens⁵⁰⁻⁵³ and previous studies on graphene-like materials⁵⁴, we assessed the shielding properties in the range 0.2-2 THz, across one frequency decade⁵⁵.”

Comment 1. The measurement set up for the assessment of the SE at THz should be better described. It should be specified which is the EM field configuration considered for the measurement of the SE (i.e. plane wave? Normal incidence?), which the calibration configuration considered, how the effect of edge and wedge scattering is considered or corrected.

Authors' reply

We thank the Reviewer for his/her comment. In the revised version we have improved and expanded the overall description of the THz SE measurement setup by adding several new details on beam properties, field configuration, mitigation of scattering effects, and calibration method. The specific paragraph in the Experimental Section now reads:

“The EMI shielding response of the nanolaminates was investigated in the range 0.2-2 THz by using a time domain spectrometer (TeraK15 from Menlo Systems) driven by a femtosecond laser fibre-coupled to photoconductive antennas both for THz emission and detection. Data acquisition was realized by means of a lock-in amplifier coupled with electronics and computer software. A standard setup with four polymethyleneterephthalate (TPX) lenses was used to collimate and focus the beam first impinging onto the sample plane and then received by the detector. Computer aided motion controller was utilized for accurate target positioning. All samples have been tested under dry conditions (< 0.1% relative humidity) in a nitrogen environment to avoid spurious effects caused by moisture. Samples are placed on an Al sample holder having a hole of radius 8 mm. Since the THz beam waist w_0 at the focus point is approximately 1.5 mm, the area under illumination is at most only 10% of the sample effective surface, making negligible any scattering given by edge or wedge effects. Since $(k_0 w_0)^2$ is much larger than unity in the investigated frequency range (k_0 being the wavevector in free space) we can safely assume that the plane wave approximation holds for all measurements that have been carried out within a few percent relative error. The THz

beam impinges on the sample surface under normal incidence. The holder is placed on a kinematic mount for fine adjustment and the sample surface parallelism with respect to the beam wavefront is controlled by a three-points calibration method. Therefore, paraxial error is considered to be extremely small. The electric field versus time was acquired separately upon transmitting through the sample ($\hat{E}_{\text{smp}}(\omega)$) and through the free space used as a reference signal ($\hat{E}_{\text{ref}}(\omega)$). Time domain signals, averaged over 3000 waveforms, are recorded over a relatively large interval up to 300 ps and then converted into the frequency domain by applying a Fast Fourier Transform (FFT). From here the electric-field ratio, the transmittance $T(\omega) = \left| \frac{\hat{E}_{\text{smp}}(\omega)}{\hat{E}_{\text{ref}}(\omega)} \right|$, was measured with a resolution better than 5 GHz.”

Comment 2. It is demonstrated that the produced Gr/PMMA nanolaminates demonstrate a substantial increase in Young’s modulus as a function of graphene content and, for the first time, it is shown that there is a linear relationship between modulus and volume fraction. A prediction model should be used to validate this result.

Authors’ reply

We thank the Reviewer for this comment. Actually, the discovery of graphene had raised major expectations for the fabrication of high-performance nanocomposites in the latest fifteen years, and one of the earliest concerns was whether the traditional composite theories could be applied to the mechanical properties of graphene-based nanocomposites. Interestingly, it was experimentally demonstrated that continuum mechanics can describe the mechanical reinforcement by graphene¹. In the micromechanics approach, the main proposed strategies to predict the effective properties of a composite material are based on the implementation of analytical methods (such as the Rule of Mixtures²) or semi-empirical methods (such as the Halpin-Tsai model³).

Actually, one of the simplest relationships that has been developed to describe the reinforcement achieved from a high-modulus filler in a low-modulus matrix, under uniform strain, is the so-called “rule of mixtures” (RoM), in which the Young's modulus of a composite E_c along the fibres direction is given by

$$E_c = E_m(1 - V_f) + V_f E_f \quad \text{Eq. R1}$$

where E_m and E_f are respectively the modulus of the matrix and of the filler, and V_m and V_f are the volume fraction of the matrix and of the filler. The RoM was developed for continuous and unidirectional fibres and its simplicity is coming from hypothetical assumptions such as the unidirectional alignment of the fibres, their infinite length and the perfect bonding between the components.

Another common approach used for nanocomposites with parallel-aligned nanoplatelets is the Halpin-Tsai model (HT). Accordingly, for nanocomposites having fillers with high

orientation degree, the Young's modulus in the direction longitudinal to the filler orientation (E_{II}) can be estimated by:

$$E_{II} = \left[\frac{1+2a\eta_{II}V_f}{1-\eta_{II}V_f} \right] E_m \quad \text{Eq. R2}$$

where, η_{II} is defined as

$$\eta_{II} = \left[\frac{\frac{E_f}{E_m} - 1}{\frac{E_f}{E_m} + 2a} \right] \quad \text{Eq. R3}$$

and a is the aspect ratio of the filler (width/thickness).

It is interesting noting that, when $a \rightarrow \infty$ which is the case of very large filler platelets, Eq. R2 is then reduced to the RoM and gives the maximum reinforcement:

$$E_{II} = E_f V_f + E_m (1 - V_f) \quad \text{Eq. R2-i}$$

We implemented the HT model to predict the Young's modulus of a PMMA-graphene composite with fully aligned layers, with different aspect ratios, a . Results are plotted in Figure R1 as a function of graphene volume fraction and are compared to the prediction based on the RoM. In the models, E_f and E_m have been set, respectively, as 1.01 TPa and 1.75 GPa. The former is modulus of pristine, defect-free graphene⁴ and the latter is the modulus of PMMA as evaluated on the bases of our tensile tests. It is clear that the reinforcement approaches the maximum reinforcement when $a > 10,000$, which is the point at which the HT model approaches the RoM.

Figure R1. Theoretical predictions of the elastic modulus of a graphene/PMMA nanocomposite (E_c) versus graphene volume fractions V_f : Halpin-Tsai (solid lines) plotted for graphene aspect ratios a and Rule of Mixture (dashed line).

For the case at hand, the aspect ratio of the proposed nanolaminates is higher than 1,000,000, therefore the RoM can be safely adopted to validate our results. Actually, in order to derive the effective modulus of graphene in the nanolaminate system, the RoM can be rewritten as follows:

$$E_c = E_m(1 - V_f) + V_f E_{Gf} \Rightarrow E_c = E_m - E_m V_f + V_f E_f$$

$$\Rightarrow E_c = E_m + V_f(E_f - E_m) \quad \text{Eq. R1-i}$$

Therefore, E_f can be derived by applying a linear square fitting to the experimental data E_c plotted as a function of V_f . In our case, by using this approach, an effective modulus for CVD graphene of 846 GPa has been estimated.

Finally, we would also point out that, as shown by some authoritative reviews in the field^{5,6}, discontinuous graphene nanocomposites hardly withstand the linear predictions of the RoM when containing graphene volume fractions higher than only 0.5-1%, with experimental properties much lower than what expected, and this is mainly due to the inefficient stress-transfer for small flakes and possible aggregation issues particularly for high graphene content, which cannot be encountered in the nanolaminate configuration, thanks to its large aspect ratio and the well-controlled continuous layered structure.

In order to take into account this comment, we have included the description of the theoretical models and the predictions in the SI, and the mathematical treatment used to derive the effective modulus of graphene.

Comment 3. The performance of the shield is described using the parameter SSE, which however does not have a physical or technical limit. In the electromagnetic compatibility community, a parameter which is considered in order to predict the minimum shielding performance of a shield is the limit for frequency attending to zero of the SE, which is directly expressed in terms of the sheet resistance of the screen. It is highly recommended that the authors use this model.

The paper may influence thinking in the field, since it present a model shielding material. However, the part related to the design of the panel is completely missing. For this reason, a comparison between modelling and experimental testing should be included in order to validate the obtained results.

Authors' reply

We thank the reviewer for his/her suggestion, since it allows us to directly compare our experimental results with the predictions of a standard model. Indeed, in a homogeneous material and for electrically thin samples at low frequencies, where $t \ll \delta$ (skin depth), and assuming a good conductor approximation, it can be shown⁷ that the total shielding effectiveness becomes frequency independent and can be expressed as:

$$SE_T = 20 \log_{10} \left(1 + \frac{Z_0}{2} \sigma t \right) \quad \text{Eq. R4}$$

where σt is the inverse of the sample sheet resistance and Z_0 is the free-space impedance.

This formula can be easily extended in the case of graphene multilayered samples introducing the sheet number N (>1) instead of the sample thickness as parameter⁸.

Values extracted using this simple model allows to predict the zero-frequency shielding performance of our samples. The numerical results have been reported in Table R1 and nicely match the extrapolated values deduced from the experimental SE_T curves as a function of frequency.

Table R1 Comparison between minimum (theoretically calculated) and measured EMI Shielding Effectiveness.

V_{Gr} [%]	σ [S/cm]	Minimum shielding [dB]	SE@0.5 THZ [dB]
0.04	4.6	3.1	2.9
0.06	6.9	4.0	7.4
0.13	8.7	5.2	10.1
0.22	16.7	8.2	12.4
0.33	28.7	11.4	15.3
0.5	93.1	21.9	23.2

The SE_T frequency dependence also corresponds to what expected taking into account the decomposition model of shielding. A thorough discussion on the role played by the different terms in the SE_T decomposition formula (eq. S5 in the SI) and a comparison between model expectations and experimental results has been included in the Supplementary Information. In particular, the following text has been added to the revised manuscript on pages 8-9, lines 25,1-2:

“The observed increase of EMI SE well compares with what theoretically expected in the limit of zero frequency (see related discussion in SI).”

Reviewer #2

In this work, the authors proposed a kind of method to fabricate centimeter-scale CVD graphene (Gr)/PMMA nanolaminates, and investigated the effect of different volume fractions of graphene on mechanical, electrical and electromagnetic properties. The maximal EMI SE reaches 60 dB at a thickness of 33 μm . This work is meaningful. Therefore, I recommend it to be published in Nature Communications after some revision.

Comment 1. The authors fabricated Gr/PMMA layers with different volume fractions of graphene and demonstrated the great effect of volume fraction on mechanical, electrical and electromagnetic properties, but there are in the lack of detailed description to fabricate Gr/PMMA layers with different volume fractions and to measure the volume fractions. This makes me confused.

Authors' reply

We thank the Reviewer for this comment. We would like to recall here that, in the nanolaminate configuration, the repetitive unit cell is a ply consisting of a monolayer graphene sheet and a PMMA film, as shown in the sketch depicted below.

Figure R1. Sketch of the nanolaminate with exploded view of the repetitive unit cell 'Gr/PMMA ply'.

Therefore, the volume fraction of graphene in the nanolaminate (V_{Gr}) is defined by the following equation:

$$V_G = \frac{t_{Gr}}{t_{Gr} + t_{PMMA}} \quad \text{Eq. R2-i}$$

where t_{Gr} is the thickness of a monolayer graphene (0.33 nm) and t_{PMMA} is the thickness of the PMMA film. In light of eq. R2-i, it is clear that, as t_{Gr} is fixed, graphene volume fraction can be increased by reducing t_{PMMA} in the repetitive unit cell. Specifically, in our work,

nanolaminates with V_{Gr} from 0.04% to 1% have been produced, and this was achieved by modulating the thickness of the polymeric film produced via spin coating, with t_{PMMA} ranging from 750 to 33 nm (Figure R2). Actually, prior starting the iterative lift-on/ float off process, spin coating was fine-tuned to produce PMMA films on graphene with set thicknesses. In this preliminary ‘thickness optimization’ phase, different PMMA solutions in anisole were employed and the angular speed was varied during spin coating: therefore ultrathin PMMA films were produced on monolayer graphene grown on Cu film. Subsequently, having etched away the Cu, the Gr/PMMA ply was deposited on a Si wafer using the transfer device shown in Figure 1 of the manuscript. The thickness of the single film was then measured by AFM according to the scratch step method⁹. In this way, we obtained the correlation between the spin coating conditions (concentration of the solution and RPM) and the thickness of the single layer (which was already reported in the Table S1), which was adopted as reference for the fabrication of nanolaminates with different graphene volume fractions. The information on the spin coating process employed to produce the ultrathin films, along with the correspondence between the volume fraction of graphene in the nanolaminate and the thickness of the polymer in the Gr/PMMA ply, was already reported in the Table S1 of the submitted paper. It is important to note here that, for each nanolaminate, the number of plies and therefore the final thickness (between 4.5 and 6.5 μm) was determined in order to guarantee a safe handling of the produced membranes during characterization. For instance, the nanolaminate with $V_{Gr} = 0.04\%$ was constructed with six Gr/PMMA plies (or layers), the thickness of each being 750 nm. The nanolaminate with $V_{Gr} = 0.5\%$ was built by depositing a hundred of 65 nm-thick Gr/PMMA plies, as well. Full details on the different nanolaminates we have produced are reported in Table R1. Finally we would like to add that the volume fraction of graphene was also inferred inversely from the thickness of each nanolaminate as measured by a digital micro-meter.

Figure R2. Relation between the thickness of the polymeric layer and the volume fraction of graphene in the nanolaminate.

Table R1. Produced CVD Gr/PMMA nanolaminates (*1 vol% model specimen is not a freestanding membrane but is supported on a quartz substrate)

Solution in Anisole (wt%)	Angular speed (RPM)	Layer Thickness (nm)	Nominal Graphene Volume Fraction (%)	N. of layers	Nominal Nanolaminate thickness (μm)	Measured Final thickness (μm)	Actual Graphene Volume Fraction (%)
3	1500	250	0	20	5	5.05	0
6	1000	750	0.044	6	4.5	4.64	0.04
6	2000	500	0.066	10	5	5.15	0.06
3	1500	250	0.13	20	5	4.98	0.13
3	2000	200	0.165	20	4	4.1	0.17
3	3000	150	0.22	30	4.5	4.58	0.22
2	1500	100	0.33	50	5	5.08	0.33
2	3000	65	0.5	100	6.5	6.47	0.5
1	1500	33	1*	4	0.132	0.132	1

Therefore, in order to clarify the way we produced nanolaminates with different volume fractions, details on the calculation of the V_{Gr} have been added in the Experimental Section, Figure R2 has been included in the SI as Figure S1 and Table S1 has been updated as shown in Table R1:

“The production of each PMMA/Gr layer was done by spin coating a PMMA solution in anisole (495 PMMA, Microchem) on the graphene grown on the copper foil of dimensions 35 mm \times 35 mm. This step is fundamental for the control of the graphene volume fraction (V_{Gr}) in the nanolaminate. In fact, V_{Gr} is defined as $\frac{t_{Gr}}{t_{Gr}+t_{PMMA}}$, being t_{Gr} the thickness of monolayer graphene (0.33 nm⁶⁶) and t_{PMMA} the thickness of the PMMA layer. The spinning conditions and concentration of the solution were preliminarily optimized to fabricate films with specific thickness, compatible with the desired graphene content in the nanolaminate (see Table S1 and Figure S1). In this optimization process, the thickness of the single PMMA/graphene layer deposited on a Si wafer was measured by using AFM according to the scratch step method.”

Comment 2. The crystal structure of single Gr/PMMA is different from that of 10 Gr/PMMA. And the authors said that the ruptures and tears will occurs in graphene when the number of depositions increases. Why? What causes the phenomena?

Authors' reply

We thank the reviewer for this comment. As mentioned earlier, the production of nanolaminates with increasing graphene volume fraction requires multiple graphene depositions in order to achieve a final thickness that can be suitable for handling. This means that the lift-off/ float-on process must be repeated on the same ‘growing’ laminate even 50-100 times or more. However, the process of multiple deposition, can introduce some structural defects in the nanolaminate. This in fact, has been verified by Raman spectroscopy which shows that, upon multiple depositions of Gr/PMMA layers, a very small increase of the I(D)/I(G) ratio vis-à-vis a single Gr/PMMA layer is indeed observed.

Figure R4. Comparison of I(D)/I(G) for 1Gr/PMMA (left) and 10 Gr/PMMA layers (right).

In order to clarify this aspect, we have included the histograms of the I(D)/I(G) ratios in the Figure 2 of the revised manuscript and we have added the following text:

“A slight increase of the I(D)/I(G) ratio in the Gr/PMMA nanolaminates indicates that structural defects are in fact induced by the lamination process.”

Furthermore, we have recently deduced that the main factor responsible for the embrittlement of the nanolaminates with the increase of graphene volume fraction (i.e. the decrease of the thickness of the single PMMA layer) over 0.20% (figure 3a) is the gradual introduction of structural defects in the polymer layer that conformally reproduces the morphology of the rough copper substrate. As shown in Figure R5a and b, the surface of the commercial Cu foil adopted for graphene growth as sacrificial substrate for the fabrication of each graphene/polymer layer, presents a roughness of around 70 nm, as revealed by AFM measurements, and in agreement with literature studies^{10,11}. The polymer layer reproduces conformally the irregular surface of the Cu substrate and, as its thickness decreases (e.g. to 65 nm for 0.5 vol%), the presence of such morphological defects contributes to the premature failure of the entire nanolaminate under tensile loading.

Figure R5. (a) Optical micrograph and (b), AFM images of copper foil after CVD process. (c) Optical micrograph of Gr/PMMA single layer deposited on Si wafer.

To verify the above, we have made some preliminary experiments on nanolaminates that have been produced by using copper foils with different roughness and varying graphene content. The stress-strain curves of these Gr/PMMA laminates are depicted in Figure R6 and clearly show that for the specimens with lower graphene content ($V_{Gr}=0.13\%$ \rightarrow $t_{PMMA}\sim 250$ nm), the mechanical behaviour is not affected by the roughness of the substrate; however, when V_{Gr} is increased to 0.28 ($t_{PMMA}\sim 120$ nm), the elongation at break (and the tensile strength) of the specimen produced using a smoother Cu foil is significantly higher. Therefore, we conclude that the roughness of the sacrificial Cu substrate affects significantly the tensile strength of the laminates particularly when the polymer layer thickness decreases.

Figure R6. Effect of roughness of Cu foil on the mechanical behaviour of the nanolaminates: Stress-strain curve of Gr/PMMA laminates produced on copper foils of different roughness. It is interesting noting that roughness of the Cu sacrificial substrate plays an increasing role as the thickness of the polymeric layer decreases. In fact, for the nanolaminates with 0.28% graphene (thickness of PMMA around 120 nm), a visible improvement of the mechanical performance is observed, while in the nanolaminates with 0.13 % (thickness of PMMA 250 nm), the effect of copper roughness is negligible on the final performance of the nanolaminate.

In order to take into account this comment, Figures R5 and R6 have been added to the SI and the following text has been added to the revised manuscript:

“Nevertheless, it is worth adding that, for graphene content over 0.22 vol%, an embrittlement of the nanolaminates is observed which could be ascribed to defects originated at random locations in the non-uniform thickness of the ultra-thin polymer/graphene layers³². Actually, optical microscopy and AFM images (Figure S3) reveal that the single polymer/graphene layer conformally reproduces the morphology of the Cu foil after CVD graphene growth, which presents typical roughness values around 70 nm or even more^{36,37}. As the thickness of the polymer layer decreases (e.g. to 65 nm for $V_g=0.5$ vol%), the contribution of such morphological defects contributes to the premature failure of the entire nanolaminate under tensile loading. Furthermore, the increase number of depositions in the lamination procedure (up to 100 in the laminate with 0.5 vol% graphene) may also introduce ruptures and tears in graphene that can affect its strength, in agreement with Raman spectroscopy. Production of nanolaminates by using higher quality graphene grown on Cu

foil with lower roughness is in progress and is expected to improve this behaviour, as shown by preliminary tests reported in Figure S4.”

Comment 3. Authors said that the thin polymer layers act as spacers whereas the graphene monolayers tend to conserve the intrinsic properties of isolated graphene, additively contributing to the overall in-plane electrical conductivity of the laminates. What is the meaning of " additively contributing to the overall in-plane electrical conductivity of the laminates"? Please explain it in detail, and analyze the reason.

Authors' reply

We thank the reviewer for his/her comment. Actually, the graphene-polymer nanolaminate consists of n alternated conductive and dielectric layers. In the ideal laminated structure, only graphene sheets contribute to the in-plane electrical conduction and the polymer layers act as insulating spacers. Therefore in this ideal composite architecture of perfectly oriented graphene layers, each of them should conserve its intrinsic properties, and the material can be modelled as a parallel system of conducting sheets with an equivalent conductance $G_{Laminate}$ equal to n times that of an isolated graphene layer G_{Gr} ^{12,13},

$$G_{Laminate} = nG_{Gr} \quad \text{Eq. 2}$$

Considering the relation of conductance G and conductivity σ , Eq. 2 leads to

$$\sigma_{Laminate} = V_{Gr}\sigma_{Gr} \quad \text{Eq. 3}$$

As shown in Figure 4a, the in-plane conductivity of the nanolaminate increase linearly with graphene volume fraction up to $V_{Gr} = 0.33\%$, and by using Eq.3, the contribution of graphene to the electrical conductivity has been evaluated from linear fitting to be $8.1 \cdot 10^3$ S/cm. The fact that this value is not far from reported values for CVD monolayer graphene transferred on Si wafer¹⁴ confirms that each graphene layer retains its intrinsic properties.

In order to clarify this aspect, the derivation of the equivalent conductivity of the laminate has been added in the SI and the amended manuscript now reads:

“In fact, unlike particular-filled composites in which electrical conductivity is limited by electron hopping^{38,40-42}, the electrical conduction in the proposed nanolaminates is due to fast electron transport in the continuous π bonds of the graphene sheets that are separated by the continuous polymer layers. Hence, in this laminated architecture of perfectly oriented graphene layers, each of them should retain its intrinsic properties, and the material can be modelled as a 2D parallel system of conducting sheets with an equivalent sheet conductance N times that of an isolated graphene layer⁴³.”

Comment 4. This is a special alternant nanolaminate structure. I suggest the authors could provide the SER and SEA to further analyze the response mechanism.

Authors' reply

We thank the reviewer for his/her suggestion. A detailed discussion on the mechanisms underlying the response for each term was already reported in the SI and has been expanded in the new version (see also our reply to comment #11 for further details). As an example, in the figure below we plot the total shielding effectiveness for the 100 layers sample (0.5 vol%) and its decomposition in the three different contributions SE_M , SE_R and SE_A (multiple internal reflections, surface reflection, and absorption respectively) in the frequency interval of investigation.

Figure R7. Frequency dependence of the total (measured) shielding effectiveness SE_{TOT} (square symbols) and its decomposition in surface reflection, internal reflections and absorption contributions for 100 layers of graphene/PMMA stack with 0.5 vol% of graphene. SE_R , SE_A and SE_M (blue, red, and green continuous lines respectively) were calculated using Eq. S1 and S2.

Comment 5. The background of the manuscript should be further highlighted over the reported literatures. Some important and relevant literatures should not be ignored, such as

Nat. Commun., 2015, 6, 6628; Adv. Mater. 2020, 2002112; Carbon 2009, 47, 922-925; Carbon, 2019, 155, 232-242.

Authors' reply

We have updated the reference list accordingly.

Comment 6. Some experiments and literature have proved that graphene/PMMA nanolaminates have better mechanical properties, but this manuscript lacks theoretical analysis or mechanism elaboration on mechanical properties enhancement.

Authors' reply

We thank the reviewer for this comment. Actually, the discovery of graphene had raised major expectations for the fabrication of high-performance nanocomposites in the latest fifteen years, and one of the earliest concerns was whether the traditional composite theories could be applied to the mechanical properties of graphene-based nanocomposites. Interestingly, it was experimentally demonstrated that continuum mechanics can describe the mechanical reinforcement by graphene¹. In the micromechanics approach, the main proposed strategies to predict the effective properties of a composite material are based on the implementation of analytical methods (such as the Rule of Mixtures²) or semi-empirical methods (such as the Halpin-Tsai model³).

Actually, one of the simplest relationships that has been developed to describe the reinforcement achieved from a high-modulus filler in a low-modulus matrix, under uniform strain, is the so-called “rule of mixtures” (RoM), in which the Young's modulus of a composite E_c along the fibres direction is given by

$$E_c = E_m(1 - V_f) + V_f E_f \quad \text{Eq. R1}$$

where E_m and E_f are respectively the modulus of the matrix and of the filler, and V_m and V_f are the volume fraction of the matrix and of the filler. The RoM was developed for continuous and unidirectional fibres and its simplicity is coming from hypothetical assumptions such as the unidirectional alignment of the fibres, their infinite length and the perfect bonding between the components.

Another common approach used for nanocomposites with parallel-aligned nanoplatelets is the Halpin-Tsai model (HT). Accordingly, for nanocomposites having fillers with high orientation degree, the Young's modulus in the direction longitudinal to the filler orientation (E_{II}) can be estimated by:

$$E_{II} = \left[\frac{1+2a\eta_{II}V_f}{1-\eta_{II}V_f} \right] E_m \quad \text{Eq. R2}$$

where, η_{II} is defined as

$$\eta_{II} = \left[\frac{\frac{E_f}{E_m} - 1}{\frac{E_f}{E_m} + 2a} \right] \quad \text{Eq. R3}$$

and a is the aspect ratio of the filler (width/thickness).

It is interesting noting that, when $a \rightarrow \infty$ (which is the case of very large filler platelets), Eq. R2 is then reduced to the RoM and gives the maximum reinforcement:

$$E_{II} = E_f V_f + E_m (1 - V_f) \quad \text{Eq. R2-i}$$

We implemented the HT model to predict the Young's modulus of a PMMA-graphene composite with fully aligned layers, with different aspect ratios, a . Results are plotted in Figure R1 as a function of graphene volume fraction and are compared to the prediction based on the RoM. In the models, E_f and E_m have been set, respectively, as 1.01 TPa and 1.75 GPa. The former is modulus of pristine, defect-free graphene⁴ and the latter is the modulus of PMMA as evaluated on the bases of our tensile tests. It is clear that the reinforcement approaches the maximum reinforcement when $a > 10,000$, which is the point at which the HT model approaches the RoM.

Figure R1. Theoretical predictions of the elastic modulus of a graphene/PMMA nanocomposite (E_c) versus graphene volume fractions V_f : Halpin-Tsai (solid lines) plotted for graphene aspect ratios a and Rule of Mixture (dashed line).

Finally, we would also point out that, as shown by some authoritative reviews in the field^{5,6}, discontinuous graphene nanocomposites hardly withstand the linear predictions of the RoM when containing graphene volume fractions higher than only 0.5-1%, with experimental properties much lower than what expected, and this is mainly due to the inefficient stress-transfer for small flakes and possible aggregation issues particularly for high graphene

content, which cannot be encountered in the nanolaminate configuration, thanks to its large aspect ratio and the well-controlled continuous layered structure.

In order to take into account this comment, we have included the description of the theoretical models and the predictions in the SI.

Comment 7. In the manuscript, the authors said "..... by using the rule of mixture, an effective elastic modulus of 846 GPa.....". The calculation process should be added, which can be put in Supporting Materials.

Authors' reply

We thank the reviewer for this comment. The aspect ratio of the proposed nanolaminates is higher than 1,000,000, therefore the RoM can be safely adopted to validate our results, as mentioned in the reply to Comment 6. Actually, in order to derive the effective modulus of graphene in the nanolaminate system, the RoM can be rewritten as follows:

$$\begin{aligned} E_c = E_m(1 - V_f) + V_f E_{Gf} &\Rightarrow E_c = E_m - E_m V_f + V_f E_f \\ &\Rightarrow E_c = E_m + V_f(E_f - E_m) \end{aligned} \quad \text{Eq. R1-i}$$

Therefore, E_f can be derived by applying a linear square fitting to the experimental data E_c plotted as a function of V_f . In our case, by using this approach, an effective modulus for CVD graphene of 846 GPa has been estimated.

In order to clarify the calculation process, we have included mathematical treatment used to derive the effective modulus of graphene in the SI.

Comment 8. Compared with graphene/polymer composites, the conductivity of the graphene/PMMA nanolaminates is raised by several orders of magnitudes. But in the manuscript, the authors just briefly attributed the enhancement to graphene's distribution state in the polymer, without involving the essence of electron transport. This is not sufficient to raise wide readers' interest. Please provide a further analysis. Actually, there are several important models that can be used to explain. For example, according to the two important models of "Electron-Hopping (EHP)" and "Aggregative-induced-charge transport (AICT)" [Carbon 2010, 48, 788-796; Carbon 2013, 65, 124-139; Adv. Mater. 2014, 26, 3484-3489.], this phenomenon can be well explained that the conductivity of dispersed graphene system is limited by electron hopping, while that of graphene/PMMA nanolaminates is promoted by fast electron migrating due to the continuous pi bond.

Authors' reply

We thank the reviewer for his/her comments and suggestions. We agree with the fact that, unlike particular-filled composites in which electrical conductivity is limited by electron hopping, the electrical conduction in the proposed nanolaminates is due to fast electron

migrating in the continuous π bonds of the graphene sheets that are separated by the continuous polymer layers. In order to include this comment in the amended version of the manuscript, we have added the following statement:

“In fact, unlike particular-filled composites in which electrical conductivity is limited by electron hopping^{38,40-42}, the electrical conduction in the proposed nanolaminates is enhanced thanks to the fast electron transport in the continuous π bonds of the graphene sheets that are separated by the homogeneously-thick and continuous insulating polymer layers.”

Comment 9. In this work, the absolute shielding effectiveness (SSET) is used to compare the EMI SE of graphene/PMMA at different thicknesses. However, SET is also relevant to thickness ($e\text{-}\alpha d$), so that the usage the SSET seems to be not appropriate as the assessment standard.

Authors' reply

We thank the reviewer for this comment that allow us to clarify our choice to use the SSE_t in the comparison of the shielding behaviour of the proposed nanolaminates to other materials. Generally, SE_T alone is not a sufficient parameter for understanding material shielding properties, as a higher shielding effectiveness can simply be achieved at a larger thickness, which directly increases the weight of the final product. Lightweight materials are a necessity for applications like aerospace and automotive industry. Therefore, when considering a material's shielding properties, mass density must be also taken into account. Therefore, a more realistic parameter is to divide SE_T by the material thickness. SSE_t has been widely adopted in the majority of the recent literature in the field as a standard to assess and compare the shielding performance of different materials^{15,16}. In light of that, we embraced this consolidated approach and used SSE_T as a criterion to evaluate the effectiveness of different materials since it incorporates three important factors at once: EMI SE_T , thickness, and density.

Comment 10. In this work, authors found that the linearity of in-plane conductivity with volume fraction is lost for higher graphene content, and attribute it to the occurrence of further mechanisms, contributing to electrical conduction. This new mechanism may be the creation of interconnection or bridging between adjacent layers. How does the creation of interconnection or bridging between adjacent layers cause such changes in conductance? Can authors give a clear explanation and provide some relevant literatures?

Authors' reply

As stated before, in the ideal alternant-layered structure, electrical conduction is due to fast electron migrating in the continuous π bonds of the graphene sheets that are separated by the continuous polymer layers. This is confirmed by the linearity of conductivity vs. volume fraction plot up to $V_{Gr} = 0.33\%$ (compatible with $t_{PMMA} \sim 150$ nm). At higher volume fraction

linearity is lost, therefore additional mechanisms contribute to electrical conduction. As stated in the reply to Comment 2, we have observed the presence of defects in the non-uniform thickness of the polymer layer that conformally reproduces the morphology of the rough copper foil. High volume fractions correspond to $t_{\text{PMMA}} < 100$ nm, which are values comparable or lower than the roughness of the Cu foil. Therefore it is likely that the inhomogeneity of the thickness of the PMMA layer can be responsible for the formation of further conductive paths through bridging and interconnection between contiguous graphene layers. As shown in the literature for other composites with highly aligned graphene layers^{17,18}, it is possible that hopping electrons enhance the micro-current in the layered package of graphene when the local thickness of the PMMA spacer is small enough (<5 nm) thus triggering through-plane conduction^{19,20,21}.

In order to take into account this comment, we have added the following text in the amended version of the manuscript:

“However, we observe that the linearity of in-plane conductivity with volume fraction is lost for higher graphene content (i.e. for $t_{\text{PMMA}} < 100$ nm), which may suggest the occurrence of further mechanisms contributing to charge transport. Actually, the aforementioned inhomogeneity of the polymeric layer thickness due to the use of a rough sacrificial substrate probably causes interconnection or bridging between adjacent layers. At this stage, therefore, it is possible that hopping electrons enhance the micro-current in the layered package of graphene when the local thickness of the PMMA spacer is small enough (<5 nm) thus triggering through-plane conduction. The conduction mechanism in this stage could be therefore ascribed to the combination of migrating and hopping electrons^{38,40,41,46,47}”

Comment 11. According to the equation S1 and S2, the SE increase with frequency. What physical mechanism inside the graphene/PMMA nano-laminates causes this change?

Authors' reply

In the transmission line model of shielding, the SE equation can be described using the so-called Schelkunoff decomposition²² as the sum of three terms, namely SA (absorption), SR (surface reflection), and SM (multiple reflections). Actually, the third term is always negative in layered systems but can be neglected as far as material losses are consistent, which is always the case but at the smallest layer numbers. Therefore, the SE behaviour as a function of frequency depends on the interplay between the first two terms only. Indeed, the SA term obviously increases with frequency since it inversely depends on the penetration depth, whereas the SR term shows a decreasing frequency behaviour as a consequence of being a “mismatch” loss: the higher the impedance difference at the interface between vacuum and graphene-polymer nanolaminates the lower the surface reflection. For the majority of samples, this implies that the two contributions balance out and the overall shielding is almost constant with frequency. However, in the limit of a good conductor approximation the

electric field is actually “shorted out” at the first interface, and the SE_R term relative weight in the SE formula diminishes²³. This is clearly seen in Fig. 4c for specimens having the same graphene volume (0.33%) but different total thickness: when absorption becomes the dominant shielding mechanism, SE as a consequence increases with frequency. We tried to better clarify the role played by the different terms in the SE decomposition formula expanding the discussion in the SI part and adding the following text:

“The second term, SE_M , expressing attenuation due to multiple internal reflections, in layered systems is actually a negative term that reduces the total shielding effectiveness. When SE_A is larger than 10 dB, usually this contribution can be neglected, which is always the case in our specimens but at the smallest layer numbers. Therefore, the SE behaviour as a function of frequency depends on the interplay between the contributions to the shielding given by material absorption and interface reflection only. Indeed, the SE_A term obviously increases with frequency since it inversely depends on the penetration depth, whereas the SE_R term shows a decreasing frequency behaviour according to the transmission line model of shielding. For the majority of samples, this implies that the two contributions balance out and the overall shielding is almost constant with frequency. However, in the limit of a good conductor approximation the electric field is actually “shorted out” at the first interface, and the SE_R term relative weight in eq. S1 diminishes. This is clearly seen in Fig. 4c for specimens having the same graphene volume (0.33%) but different total thickness: when absorption becomes the dominant shielding mechanism, SE as a consequence increases with frequency. Moreover, as expected from eq. S2, the higher the conductivity the larger is the contribution given by SE_A . Nevertheless, a further increase in overall conductivity produced by reducing thickness of the single graphene/PMMA layer will produce a minor effect on the total shielding effectiveness since this term linearly depends on t .”

Reviewer #3

Recommendation: Reject

This paper by Pavlou et al reports the synthesis/assembly of graphene/PMA nanolaminates via a 'lift-off/ float-on' process for THz EMI shielding. The nanolaminated structure reached EMI SE of 60 dB for a small thickness of 33- μm . Although the EMI SE results are impressive, the methodology of nanolaminated structure is not new and there is no significant attraction in the results which can merit publication in Nature Communications. I will suggest this article for publication in Scientific Reports, after addressing following comments.

Authors' Reply

We respectfully disagree with the Reviewer on this point. Actually, the current paper provides a substantial contribution of results, methodology and theoretical analysis to the field of EMI shielding. Moreover, our process route for fabricating all-fluidic, automated and iterative lift-off /float-on nanolaminates with maximum size of $7\times 7\text{ cm}^2$, is quite unique and differs a great deal from past attempts. In particular, with the recent development of roll-to-roll CVD graphene our fabrication process allows the manufacturing of laminates of any required size (NB currently a 1m-long nanolaminate is under preparation in our lab). In contrast the folding process reported by Ruoff et al recently²⁴ involves successive folding of a single graphene/ polymer unit which leads to nanolaminates of a few mm in length.

Comment 1. The authors have published the EMI shielding results of same work earlier in IEEE paper <https://ieeexplore.ieee.org/stamp/stamp.jsp?arnumber=8874200> in 2019. Since the shielding values are the main object of these works, the current submission provides no new insight on shielding values.

Authors' reply

We would like to point out to the Reviewer that the one page IEEE Abstract mentioned above does not constitute a scientific manuscript and on the basis of this, the statement above that the "current submission provides no new insight" is inappropriate. Furthermore, we strongly believe that a conference presentation prior to submission of a full manuscript does not constitute a violation of the rules of submission to Nature journals. In particular, the IEEE Abstract reports only preliminary results on the THz shielding of the nanolaminates, without providing any information on the production of the nanolaminates, their mechanical & electrical characterization, full EMI shielding behaviour as a function of density and thickness and, finally, theoretical analysis. In fact, the Nature Communications submission provides clearly a new insight/ paradigm over the whole concept of EMI shielding, as pointed

out also by one of the reviewers. In conclusion, the current paper proposes a novel way of nanolaminate production that yields 3D large composites, unrestricted by dimensional concerns, and introduces a novel approach for assessing EMI behaviour of material systems. It is thus of extreme interest to a wide readership and thus it warrants publication to Nature Communications.

Comment 2. Although the nanolaminated structure provides good shielding, the process is quite complex and slow as compared to other composite synthesis process, for example, development of foam structures with very low graphene filler content in polymer matrices.

Authors' reply

We thank the Reviewer for this comment that allows us to clarify and define the status of the cm-scale CVD-graphene nanolaminates both in the state of the art as novel multi-functional material and in the market as potential shielding sheets for applications where small thicknesses and low weights are required. As stated in the manuscript “*with the development of compact electronic devices, the requirements for EMI shielding materials are moving toward light-weight, flexible systems capable to exhibit strong absorption per unit volume and/or weight. It is clear that the current challenge in EMI shielding for aerospace and electronics is to achieve high EMI SE with low added weight and at small thicknesses*”. In fact, in the last years, scientific community has paid particular attention to the development of membranes with small thickness, mechanical integrity and high shielding performance^{15,16}, instead of the easy-to-fabricate but thick and fragile foams.

In our case, we synthesize CVD graphene in our labs by using the AIXTRON Black Magic reactor, which is a semi-industrial furnace that can accommodate multiple substrates or a single 4" wafer. Therefore, it is obvious that typical *batch CVD process* does limit *per se* the production of nanolaminates to squares of side ~7 cm which are obtained through a quite laborious process. Current developments for *roll-to-roll CVD production*²⁵ could allow us to use larger sheets of graphene that can be coated with ultra-thin polymer in order to produce nanolaminates with larger lateral dimension and in a continuous process. In fact, the ideal combination of roll-to-roll CVD furnaces with lamination machines could be used in future to design a viable, continuous industrial process to manufacture hundreds of meters of nanolaminates in rolls. However, we are not looking yet at mass production. At this stage, our primary intention is to disseminate to the scientific and industrial community the concept of CVD graphene nanolaminate that outperforms SOTA flake-based graphene polymer composites and is produced with a novel process which is indeed scalable. However, even in the current dimensions, the 7 cm × 7 cm nanolaminate can be applicable in several real applications thanks to the excellent shielding properties, for example as shielding sheets against undesirable electromagnetic signals in electronic enclosures, or as patches on larger composite panels for aeronautics/automotive applications. Finally, we would like to highlight that the technology of CVD graphene growth and transfer, is rapidly developing and significant advances have been made. In fact, in less than one decade, it has moved from the very first growth and transfer of CVD graphene^{26,27} to the production of a 100-m-long high-

quality graphene on transparent conductive film by roll-to-roll process. Therefore, based on the increasing annual production capacities and the resulting cheapest prices²⁸, a larger-scale and lower-cost growth and transfer of CVD graphene is expected in the nearest future, which will have a major impact on the development of new composite materials, such as our nanolaminates.

Comment 3. The results largely depend on the control of polymer thickness layer via spin coating. How this process can be scalable to meet commercial requirements?

Authors' reply

A suitable way to scale-up the production of CVD graphene/polymer nanolaminates, as mentioned in the Reply to Comment 3, is represented by the combination of roll-to-roll CVD furnaces with lamination machines. In such an automatic process, the deposition of the polymer layer could be performed via ultrasonic spray coating, which has been demonstrated to ensure thickness smaller than 500 nm^{29,30}. A preliminary design of the system is shown in the figure below. However, we want to underline here that, as stated above, at the moment our primary intention is to disseminate the concept of CVD graphene nanolaminate that outperforms SOTA flake-based graphene polymer composites and is produced with a novel process which is indeed scalable.

Figure R1. Schematic of the automatic spray coating phase for the deposition of thin polymer films on continuous graphene on Cu roll.

Comment 4. For higher graphene content linearity in electrical conductivity is lost. The authors shall state the proper reason, why in-plane electrical conductivity, in the presence of a polymer spacer tends to increase significantly at higher graphene layers? The electrical conductivity mechanisms may be explained.

Authors' reply

We thank the Reviewer for his/her comment. Actually, as reported in Refs. 19-21, the electrons can 'migrate' in one graphene layer or 'jump' across the defects or interface between layers; these are identified as the migrating and hopping electron, respectively. We can safely assume that at small volume fractions ($t_{PMMA} > 100$ nm), the polymer layer is continuous and the nanolaminate behaves as an ideal alternant-layered structure, in which only graphene sheets contribute to electrical conduction and the polymer layers act as spacers. In this ideal composite architecture of perfectly oriented graphene layers, the electrical conduction is due to fast electron migrating in the continuous π bonds of the graphene sheets that are separated by the continuous insulating polymer spacers. In light of that, each graphene layer should conserve its intrinsic properties, and the nanolaminate can be modelled as a parallel system of conducting sheets with an equivalent conductance $G_{Laminate}$ equal to n times one of an isolated graphene layer G_{Gr} ^{12,13}

$$G_{Laminate} = nG_{Gr} \quad \text{Eq. 2}$$

Considering the relation of conductance G and conductivity σ , Eq. 2 leads to

$$\sigma_{Laminate} = V_{Gr}\sigma_{Gr} \quad \text{Eq. 3}$$

As shown in Figure 4a of the manuscript, the in-plane conductivity of the nanolaminate increase linearly with graphene volume fraction up to $V_{Gr} = 0.33\%$, and by using Eq.3, the contribution of graphene to the electrical conductivity has been evaluated from linear fitting to be $8.1 \cdot 10^3$ S/cm. The fact that this value is not far from reported values for CVD monolayer graphene transferred¹⁴ on Si wafer confirms that each graphene layer retains its intrinsic properties.

At higher volume fractions, linearity of electrical conductivity is lost and this can be attributed to further mechanism of conduction. Actually, the polymer layer conformally reproduces the morphology of the copper foil. Recent AFM experiments have revealed that the surface of the Cu foil is not smooth, with roughness around 70 nm (Figure R2a and b). This finding is in agreement with other literature studies^{10,11}. Therefore, as the PMMA layer becomes thinner, defects may be progressively introduced in its non-uniform thickness, as shown in Figure c. These defects in the polymer spacer can be responsible for the formation of further conductive paths through bridging and interconnection between contiguous graphene layers. As shown in the literature for other composites with highly aligned graphene

layers¹⁷⁻¹⁸, it is possible that hopping electrons enhance the micro-current in the layered package of graphene when the local thickness of the PMMA spacer is small enough (<5 nm) thus triggering through-plane conduction¹⁹⁻²¹.

Figure R2. (a) Optical micrograph and (b), AFM images of copper foil after CVD process. (c) Optical micrograph of Gr/PMMA single layer deposited on Si wafer.

In order to include our reply to this comment, the following text has been added:

“In fact, unlike particular-filled composites in which electrical conductivity is limited by electron hopping^{38,40-42}, the electrical conduction in the proposed nanolaminates is due to fast electron transport in the continuous π bonds of the graphene sheets that are separated by the homogeneously-thick and continuous polymer layers. Hence, in this alternant-layered architecture of perfectly oriented graphene layers, each of them should retain its intrinsic properties, and the material can be modelled as a 2D parallel system of conducting sheets with an equivalent sheet conductance N times that of an isolated graphene layer⁴³.”

“However, we observe that the linearity of in-plane conductivity with volume fraction is lost for higher graphene content (i.e. for $t_{PMMA} < 100$ nm), which may suggest the occurrence of further mechanisms contributing to charge transport. Actually, the aforementioned inhomogeneity of the polymeric layer thickness produced by a rough sacrificial substrate probably causes interconnection or bridging between adjacent layers. At this stage, therefore, it is possible that hopping electrons enhance the micro-current in the layered

package of graphene when the local thickness of the PMMA spacer is small enough (<5 nm) thus triggering through-plane conduction. The conduction mechanism in this stage could be therefore ascribed to the combination of migrating and hopping electron^{38,40,41,46,47}.”

Comment 5. The process of increasing the overall thickness (Figure 4c) is not explained in experimental section.

Authors’ reply

We thank the Reviewer for his/her comment. The overall thickness of the nanolaminate, at a fixed V_G , can be modulated by varying the number of depositions. Therefore, increasing the thickness means increasing the number of deposited layers. In particular, in the case at hand, the nanolaminate had a fixed $V_G = 0.28\%$, and the three specimens consisted of 50, 100 and 250 120 nm-thin layers. As already mentioned in the manuscript, in order to increase the thickness of the nanolaminate, *“the cycle was repeated until the desired number of PMMA/Gr layers was reached.”*

Comment 6. The SEM image in Figure 1c does not reveal any information related to graphene. The layers shown are probably due to polymer spacers; how many graphene layers are expected in, for example 0.3 vol % nanolaminated structure?

Authors’ reply

As stated above, the number of layers can be modulated, according with the desired final thickness of the membrane and with the thickness of a single layer (defining the volume fraction of graphene in the nanolaminate). For instance, in order to fabricate a nanolaminate with $V_g = 0.13\%$ (i.e. the specimen depicted in the SEM image in Figure 1c), a thickness t_{PMMA} of 250 nm is required for the single PMMA layer, as the graphene volume fraction is defined as:

$$V_{Gr} = \frac{t_{Gr}}{t_{Gr} + t_{PMMA}}$$

The specific specimens shown in the SEM has a thickness of 5 μm and consists of 20 PMMA/Gr layers.

In order to clarify this point, in the amended version of the manuscript, we have added further details about the fabrication of the nanolaminates, including information on number of deposited layers and final thickness.

References

- ¹ Gong L, Kinloch IA, Young RJ, Riaz I, Jalil R, Novoselov KS (2010). Interfacial stress transfer in a graphene monolayer nanocomposite, *Adv Mater*;22:2694.
- ² Voigt W (1889). *Über die Beziehung zwischen den beiden Elastizitätskonstanten Isotroper Körper*. *Wied. Ann*, 38 573-587.
- ³ Halpin JC, Kardos JL (1976). Halpin–Tsai equations – review. *Polym Eng Sci*;16:344–52.
- ⁴ Lee, C., Wei, X., Kysar, J. W., & Hone, J. (2008). Measurement of the elastic properties and intrinsic strength of monolayer graphene. *science*, 321(5887), 385-388.
- ⁵ Young, Robert J., et al. (2012). The mechanics of graphene nanocomposites: a review. *Composites Science and Technology* 72.12: 1459-1476.
- ⁶ Papageorgiou, D. G., Kinloch, I. A., & Young, R. J. (2017). Mechanical properties of graphene and graphene-based nanocomposites. *Progress in Materials Science*, 90, 75-127.
- ⁷ P. Saini and M. Arora (2012). “Microwave absorption and EMI shielding of nanocomposites based on intrinsically conducting polymers, graphene and carbon nanotubes”, in *New Polymers for Special Applications*, Chap. 3, IntechOpen.
- ⁸ A. G. D'Aloia, M. D'Amore, and M.S. Sarto, (2015). “Optimal terahertz shielding performances of flexible multilayer screens based on chemically doped graphene on polymer substrate”, *IEEE International Symposium on Electromagnetic Compatibility*, 2015-September, art. no. 7256309, pp. 1030-1035.
- ⁹ Hong, X., Gan, Y., & Wang, Y. (2011). Facile measurement of polymer film thickness ranging from nanometer to micrometer scale using atomic force microscopy. *Surface and interface analysis*, 43(10), 1299-1303.
- ¹⁰ Procházka P, Mach J, Bischoff D, Lišková Z, Dvořák P, Vaňatka M, et al. (2014). Ultrasoother metallic foils for growth of high quality graphene by chemical vapor deposition. *Nanotechnology*, 25(18): 185601.
- ¹¹ Wu X, Zhong G, D'Arsié L, Sugime H, Esconjauregui S, Robertson AW, et al. (2016). Growth of Continuous Monolayer Graphene with Millimeter-sized Domains Using Industrially Safe Conditions, *Scientific Reports*, 6(1): 21152.

-
- ¹² K. Batrakov, P. Kuzhir, S. Maksimenko, A. Paddubskaya, S. Voronovich, P. Lambin, T. Kaplas, and Y. Svirko (2014). *Sci. Rep.* 4, 7191.
- ¹³ Lobet, M., Majerus, B., Henrard, L., & Lambin, P. (2016). Perfect electromagnetic absorption using graphene and epsilon-near-zero metamaterials. *Physical Review B*, 93(23), 235424.
- ¹⁴ Li X, Zhu Y, Cai W, Borysiak M, Han B, Chen D, et al. (2009). Transfer of Large-Area Graphene Films for High-Performance Transparent Conductive Electrodes. *Nano Letters*, 9(12): 4359-4363.
- ¹⁵ Iqbal, A., Shahzad, F., Hantanasirisakul, K., Kim, M. K., Kwon, J., Hong, J., ... & Koo, C. M. (2020). Anomalous absorption of electromagnetic waves by 2D transition metal carbonitride Ti₃CNT_x (MXene). *Science*, 369(6502), 446-450.
- ¹⁶ Choi, H. K., Lee, A., Park, M., et al. (2021). Hierarchical Porous Film with Layer-by-Layer Assembly of 2D Copper Nanosheets for Ultimate Electromagnetic Interference Shielding”, *ACS Nano*, 15, 1, 829–839, 2021
- ¹⁷ Yousefi, N., Gudarzi, M. M., Zheng, Q., Aboutalebi, S. H., Sharif, F., & Kim, J. K. (2012). Self-alignment and high electrical conductivity of ultralarge graphene oxide–polyurethane nanocomposites. *Journal of Materials Chemistry*, 22(25), 12709-12717.
- ¹⁸ Yousefi, N., Sun, X., Lin, X., Shen, X., Jia, J., Zhang, B., ... & Kim, J. K. (2014). Highly aligned graphene/polymer nanocomposites with excellent dielectric properties for high-performance electromagnetic interference shielding. *Advanced Materials*, 26(31), 5480-5487.
- ¹⁹ Wen, B., Cao, M. S., Hou, Z. L., Song, W. L., et al. (2013). Temperature dependent microwave attenuation behavior for carbon-nanotube/silica composites. *Carbon*, 65, 124-139.
- ²⁰ Cao, M. S., Song, W. L., Hou, Z. L., Wen, B., & Yuan, J. (2010). The effects of temperature and frequency on the dielectric properties, electromagnetic interference shielding and microwave-absorption of short carbon fiber/silica composites. *Carbon*, 48(3), 788-796.
- ²¹ Marsden, A. J., Papageorgiou, D. G., Vallés, C., Liscio, A., Palermo, V., et al. (2018). Electrical percolation in graphene–polymer composites. *2D Materials*, 5(3), 032003.
- ²² S. A. Schelkunoff (1934). “The impedance concept and its application to problems of reflection, refraction, shielding and power absorption,” *Bell. Syst. Tech. J.* 13, 532

-
- ²³ C. R. Paul (2006). “Introduction to Electromagnetic Compatibility”, Second Edition, Wiley & Sons, Inc..
- ²⁴ Wang, B., Li, Z., Wang, C., Signetti, S., Cunnning, B. V., Wu, X., ... & Ruoff, R. S. (2018). Folding Large Graphene-on-Polymer Films Yields Laminated Composites with Enhanced Mechanical Performance. *Advanced Materials*, 30(35), 1707449.
- ²⁵ <https://phys.org/news/2019-06-cost-effective-large-scale-graphene-aixtron.html>
- ²⁶ A. Reina, X. Jia, J. Ho, D. Nezich, H. Son, V. Bulovic, M. S. Dresselhaus, J. Kong (2009). Large area, few-layer graphene films on arbitrary substrates by chemical vapor deposition. *Nano Lett.* 9, 30–35.
- ²⁷ X. Li, W. Cai, J. An, S. Kim, J. Nah, D. Yang, R. Piner, A. Velamakanni, I. Jung, E. Tutuc, S. K. Banerjee, L. Colombo, R. S. Ruoff (2009). Large-area synthesis of high-quality and uniform graphene films on copper foils. *Science* 324, 1312–1314.
- ²⁸ Global Graphene Markets Report 2017-2018 & 2019-2030: Production Capacities - Historical, Current and Forecasts.
- ²⁹ Bose, S., Keller, S. S., Alstrøm, T. S., Boisen, A., & Almdal, K. (2013). Process optimization of ultrasonic spray coating of polymer films. *Langmuir*, 29(23), 6911-6919.
- ³⁰ Liu, S., Zhang, X., Zhang, L., & Xie, W. (2016). Ultrasonic spray coating polymer and small molecular organic film for organic light-emitting devices. *Scientific reports*, 6(1), 1-10.

REVIEWER COMMENTS

Reviewer #1 (Remarks to the Author):

We thank the authors for the detailed rebuttal.

We think that in the present form, the manuscript is still not suitable for publication on Nature Communication, since the understanding of the EM mechanism is not fully understood.

Actually, from a very simple calculation it results that a graphite layer having thickness of 33 microns and an ideal conductivity of 100 kS/m (which is equivalent to a "poor quality" graphite in the d.c. limit), provides a minimum shielding effectiveness of about 56 dB in the low frequency limit. This value takes into account only the contribution of the reflection loss. In the THz regime it has been demonstrated that absorption losses contributes to the SE, but it is also demonstrated that the a.c. conductivity of graphene decreases with the frequency, in line with Drude's model.

The value of SE of about 56 dB is in line with the value of SE measured in this work for a laminate having the total thickness of 33 microns at 2 THz, following a linear increasing trend with the frequency (see fig. 4c).

However, data from literature show that the a.c. conductivity of graphene is characterized by a steep decreasing trend with frequency higher than 1 THz.

Therefore, the experimental data reported in the manuscript need some further investigations before final acceptance for publication.

It would be necessary to clarify, for each laminate:

- 1) the total thickness of PMMA in each laminate
- 2) the thickness of the quartz substrate used in the measurements
- 3) the total thickness of graphene in the laminate

Then, it is suggested to measure the SE provided in the same frequency range by a reference material made of the quartz substrate coated with only the total amount of PMMA.

As regards the measurement configuration, it is required that the reference E-field measurement is performed considering the circular Al holder with unloaded hole. This is not specified.

The EM modelling is still not adequate.

The low-frequency approximation for SE calculation as reported in eq. 4 of the rebuttal is not adequate for prediction of the shielding effectiveness in the TH range from 0.2 to 2 THz, since it is widely demonstrated from the literature even reported yet in the reference list that the a.c. electrical conductivity of graphene decreases in the THz regime, according to Drude's law.

Data reported in "Table R1 Comparison between minimum (theoretically calculated) and measured EMI

Shielding Effectiveness" are missing on information about the thickness.

It is said that the sheet resistance is measured in d.c. using the four probe method. Since PMMA is a soft material we assume that the tip of the test fixture penetrating through the laminate create a short circuit among all layers of the laminate.

Therefore, in order to overcome this lack, we suggest that the authors compare their measurements and results with data available in the literature for very similar configurations, as reported for instance in some of the cited papers in the reference list.

Reviewer #2 (Remarks to the Author):

This revised manuscript has been carefully revised and reached the publishing standard. I recommend it to be published in Nature Communications without further revision.

Reviewer #3 (Remarks to the Author):

Nature Communications
Recommendation: Accepted

The following paper entitled "Record EMI shielding behavior of thin graphene/PMMA nanolaminates in the THz range" highlights graphene/PMMA composites synthesized by lift-off/float-on deposition process to produce multilayered laminates. In revision, authors have made substantial changes in the revised manuscript, addressing all the comments raised to a satisfactory level. The presentation of the results can be interesting for a broad scientific community of the field. Thereby, we suggest the article for its publication in the prestigious journal of Nature Communications.

Reply to Reviewers' Comments

We would like to thank Reviewers #2 and #3 for their final positive assessment of the revised manuscript and Reviewer #1 that gives us the opportunity to clarify further the mechanism of EMI shielding provided by the PMMA/graphene nanolaminates.

In the following the additional points raised by Reviewer #1 are fully addressed.

Reviewer #1

We thank the authors for the detailed rebuttal. We think that in the present form, the manuscript is still not suitable for publication on Nature Communication, since the understanding of the EM mechanism is not fully understood. Actually, from a very simple calculation it results that a graphite layer having thickness of 33 microns and an ideal conductivity of 100 kS/m (which is equivalent to a "poor quality" graphite in the d.c. limit), provides a minimum shielding effectiveness of about 56 dB in the low frequency limit. This value takes into account only the contribution of the reflection loss. In the THz regime it has been demonstrated that absorption losses contribute to the SE, but it is also demonstrated that the a.c. conductivity of graphene decreases with the frequency, in line with Drude's model.

The value of SE of about 56 dB is in line with the value of SE measured in this work for a laminate having the total thickness of 33 microns at 2 THz, following a linear increasing trend with the frequency (see fig. 4c). However, data from literature show that the a.c. conductivity of graphene is characterized by a steep decreasing trend with frequency higher than 1 THz.

Therefore, the experimental data reported in the manuscript need some further investigations before final acceptance for publication.

Authors' reply

In the following we are providing a point-to-point reply to the reviewer concerns, trying to better clarify the mechanism of the EMI shielding that governs our measurements. All changes in the main text (MT) and in the supplementary material (SM) are highlighted in blue.

As for the comparison between our 33 μm -thick Gr/PMMA nanolaminate and an equivalent graphite sample, we feel that the reviewer is somehow missing the main thesis of the paper,

that is to demonstrate that a polymer with a very tiny content of graphene (in the case of the above mentioned sample, a mere 0.33% vol.) and a multilayered architecture exhibits record EMI shielding values while keeping superior properties in terms of mechanical integrity and light weightiness (not to mention also of environmental impact). In other words the comparison of a solid piece of graphite with an alternating polymer/ graphene layered structure that contains multiple heterogeneous interfaces is somehow misleading. As for the a.c. conductivity of graphene, we are not aware of literature data showing a steep decrease at frequencies higher than 1 THz, and our experimental data show that conductivity remains flat in our experimental range (0.2 – 2 THz). A thorough discussion on this point is presented also below.

Comment a.

It would be necessary to clarify, for each laminate:

- 1) the total thickness of PMMA in each laminate
- 2) the thickness of the quartz substrate used in the measurements
- 3) the total thickness of graphene in the laminate

Authors' reply

First of all we would like to recall here that our paper deals mainly with freestanding nanolaminates with graphene volume fractions of 0.04 to 0.5%; in addition, specimens supported on quartz have been produced to assess the impressive potential of this class of material at higher content of graphene (1 vol%). This was explicitly specified in the manuscript on pages 3-4, and in the Experimental Section (page 12, line 24; page 13 lines 1-3). Furthermore, details on the calculation of the graphene volume fraction and on the thickness of the PMMA layer for each nanolaminate were already reported in the Supplementary Table S1. In particular, as stated in the Experimental Section the V_{Gr} is defined as $\frac{t_{Gr}}{t_{Gr}+t_{PMMA}}$, being t_{Gr} the thickness of monolayer graphene (0.334 nm⁶⁶) and t_{PMMA} the thickness of the PMMA layer." Since t_{Gr} is always much smaller than t_{PMMA} , then the graphene volume fraction expresses also the thickness ratio between graphene and PMMA for each specimen. Therefore the total PMMA thickness practically coincides with the final thickness of each nanolaminate (since the total Gr thickness contributes 1% max to the overall volume and therefore to the sample thickness). This can be also inferred simply multiplying in table S1 the PMMA layer thickness (column 3) by the number of layers

(column 5) for each nanolaminate. In fact, each graphene layer nominal thickness is 0.334 nm only, as explicitly stated on page 12-MT. To avoid causing confusion, we have added on page 12 the following sentence:

“However, since t_G is always much smaller than t_{PMMA} , then the graphene volume fraction expresses also the thickness ratio between graphene and PMMA for each specimen.”

As for the thickness of the quartz substrate (1 mm), this value was missing in the paper and now it has been added in the Experimental Section, where we describe the procedure for the transmittance measurements (on page 14-MT, see below).

Comment b.

Then, it is suggested to measure the SE provided in the same frequency range by a reference material made of the quartz substrate coated with only the total amount of PMMA.

Authors' reply

We thank the reviewer for pointing out this missing albeit relevant information. Indeed, this is exactly what we have done in the case of the 1 vol% nanolaminate, the only sample supported on a quartz substrate. The transmission is normalised to the bare quartz and not to the substrate coated with PMMA, since actually the polymer formally contributes to the nanolaminate overall shielding (even if for an almost unnoticeable quantity, see fig. S6-SM where we show the SE frequency dependence for a neat 5 μm PMMA film).

We have therefore modified the sentence on page 14-MT in the following way: “The electric field versus time was acquired separately upon transmitting through the sample (\hat{E}_{smp}) and through the reference material (\hat{E}_{ref} , air in case of freestanding samples, a 1 mm-thick quartz substrate in case of the 1 vol% nanolaminate)”

Comment c.

As regards the measurement configuration, it is required that the reference E-field measurement is performed considering the circular Al holder with unloaded hole. This is not specified.

Authors' reply

In order to better specify the reference E-field measurements, we have added the following sentence in the Experimental Section, page 14-MT:

“In all cases, the reference measurements were performed under the same experimental conditions applied to the samples, namely with the pulsed signal passing through the holed Al holder”.

Comment d.

The EM modelling is still not adequate. The low-frequency approximation for SE calculation as reported in eq. 4 of the rebuttal is not adequate for prediction of the shielding effectiveness in the TH range from 0.2 to 2 THz, since it is widely demonstrated from the literature even reported yet in the reference list that the a.c. electrical conductivity of graphene decreases in the THz regime, according to Drude's law.

Authors' reply

We respectfully disagree with this comment, since many available results in literature show for graphene an a.c. electrical conductivity that is still almost flat in the THz region (see, for example, Tomaino et al., Opt. Express 19, 141 (2011). [10.1364/OE.19.000141](https://doi.org/10.1364/OE.19.000141); Maeng et al., Nano Lett. 12, 551 (2012). [10.1021/nl202442b](https://doi.org/10.1021/nl202442b); Jnawali et al., Nano Lett. 13, 524 (2013). [10.1021/nl303988q](https://doi.org/10.1021/nl303988q); Zou et al., Phys. Rev. Lett. 110, 067401 (2013). [10.1103/PhysRevLett.110.067401](https://doi.org/10.1103/PhysRevLett.110.067401); Whelan et al., Opt. Express 26, 17749 (2018). [10.1364/OE.26.017748](https://doi.org/10.1364/OE.26.017748)).

This is in line with the experimental measurements performed on the 33 μm nanolaminate and shown in Fig. 4c. As we thoroughly discuss in the SM, in the shielding mechanism of this sample eq. (6) mostly rules, causing a sublinear increase of the SE as a function of frequency. In the graph below, the different contributions to the EMI SE given in eqs. S5 and S6 using a constant conductivity value are plotted. Their sum (eq. 5, dashed line) nicely matches with the experimental data, showing that (i) absorption prevails as shielding mechanism and (ii) Drude free electron model works but with a broadened spectral width of the scattering. This is likely due to the prevalence of intraband versus interband conductivity, however an in-depth discussion on this point is far beyond the scope of the present paper.

To take into account the reviewer's remark, we have added the following sentence in the SM: "It is worth to highlight that in all measurements presented here the frequency behaviour of the shielding effectiveness is consistent with a simple Drude scenario where the graphene electrical conductivity still shows a flat response in the low THz region, consistently with many literature reports [16-18]. Of course, it is possible that a frequency drop in the electrical conductivity and therefore in the SE dependence occurs at higher frequencies, in agreement with the predictions given in [6]."

Figure 1. Comparison of experimental data of shielding effectiveness (\square) and the theoretical values (dotted line) for the 33 μm -thick laminate. SE_R , SE_A and SE_M (blue, red, and green continuous lines respectively) were calculated according to eqs. S5 and S6.

Comment e.

Data reported in "Table R1 Comparison between minimum (theoretically calculated) and measured EMI Shielding Effectiveness" are missing on information about the thickness.

Authors' reply

To ease the evaluation of the EMI SE values theoretically calculated using Eq. (4), we have added in table S3-SM (Table R1 in the rebuttal letter) the thickness values already listed in table S1-SM.

Comment f.

It is said that the sheet resistance is measured in d.c. using the four probe method. Since PMMA is a soft material we assume that the tip of the test fixture penetrating through the laminate create a short circuit among all layers of the lamintate.

Authors' reply

In two recent works dealing with similar systems by Vlassioug et al. and Liu et al. (Refs. 17-18 in MT), the electrical resistance was measured in the 2-point scheme by a multimeter with conductive silver paste as electrodes at the specimens' ends. Therefore, we have adopted a similar methodology but in the 4-wire Kelvin configuration in order to minimize contact resistance. It is interesting to note that, as shown in Figure 4a, the electrical conductivity of the PMMA/Gr nanolaminates with 0.13 vol% of graphene replicates the value measured by Vlassioug et al. for the nanolaminate with the same content of graphene. This, along with the fact that the contribution of graphene to the electrical conductivity¹ is close to reported values for CVD monolayer graphene transferred on Si wafer, make us confident on the reliability of our measurements.

¹ As reported in the previous reply to reviewers and stated in the revised manuscript, at small volume fractions ($t_{\text{PMMA}} > 100$ nm), the polymer layer is continuous and the nanolaminate behaves as an ideal alternant-layered structure, in which only graphene sheets contribute to electrical conduction and the polymer layers act as spacers. In this ideal composite architecture of perfectly oriented graphene layers, the electrical conduction is due to fast electron migrating in the continuous π bonds of the graphene sheets that are separated by the continuous insulating polymer spacers. In light of that, each graphene layer should conserve its intrinsic properties, and the nanolaminate can be modelled as a parallel system of conducting sheets with an equivalent conductance G_{Laminate} equal to n times one of an isolated graphene layer G_{Gr} . The in-plane conductivity of the nanolaminate has been found to increase linearly with graphene volume fraction up to $V_{\text{Gr}} = 0.33\%$, and by considering that $\sigma_{\text{Laminate}} = V_{\text{Gr}} \sigma_{\text{Gr}}$, the contribution of graphene to the electrical conductivity has been evaluated from linear fitting. At higher volume fractions, linearity of electrical conductivity is lost and this has been attributed to further mechanism of conduction involving bridging and interconnection between contiguous graphene layers that are caused from the introduction of defects in the non-uniform thickness of PMMA layers with $t_{\text{PMMA}} < 100$ nm.

Comment g.

Therefore, in order to overcome this lack, we suggest that the authors compare their measurements and results with data available in the literature for very similar configurations, as reported for instance in some of the cited papers in the reference list.

Authors' reply

As already discussed in the reply to comment d), the SE measurements are consistent with THz conductivity data available in literature, indicating that our experimental range is well below the Drude roll-off frequency. We have added to the reference list a selection of the abovementioned literature papers:

[16] Tomaino JL, et al. (2011). Terahertz imaging and spectroscopy of large- area single-layer graphene. Optics Express 19, 141.

[17] Maeng I, et al. (2012). Gate-Controlled Nonlinear Conductivity of Dirac Fermion in Graphene Field-Effect Transistors Measured by Terahertz Time- Domain Spectroscopy. Nano Letters 12, 551.

[18] Whelan PR, et al. (2018). Conductivity mapping of graphene on polymeric films by terahertz time-domain spectroscopy. Optics Express 26, 17749.

We have also mentioned the predictions presented in [6] for a commercial graphene on PET.

REVIEWER COMMENTS

Reviewer #2 (Remarks to the Author):

1. The graphene/polymer nanolaminates with an excellent shielding effectiveness up to 60 dB at a small thickness of 33 μm , which is a very exciting. Hence, the transmission process and loss mechanism of electromagnetic waves inside should be clearly revealed to help with the application in the high-frequency field.
2. Why is it possible for jumping electrons to enhance the micro-current in the graphene layered package when the thickness of the PMMA spacer layer is less than 5 nm? How do the authors determine this thickness?
3. Author think that, for the nanolaminates with a fixed graphene volume fraction (0.33 vol.%), losses dominate the shielding mechanism. However, it is generally believed that as the thickness increases, the conductivity will increase significantly. The high-efficiency EMI shielding should depend on reflection of electromagnetic wave. Therefore, I am curious about the authors' viewpoint. Is there any evidence for this?
4. The authors give a detailed response to Reviewer #1's comments and corresponding modifications, and put forward to different viewpoint on the issue of AC conductivity (Comment d.). I think that the reviewer and the authors have a deviation in the understanding of the material object, which leads to a difference of opinion on the AC conductivity.
5. Drude's model is originated from the theory of free electron gas and is applicable to conductors such as graphene or graphene nanosheets. However, the Gr/PMMA reported in this article is composite material, and its conduction mechanism follows the jumping electron model. Therefore, the change rule of the AC conductivity of these two is not necessarily same.
6. I'm curious about the shielding performance of this graphene/polymer nanolaminate in the GHz region. Authors may choose to answer this question.

Reviewer #3 (Remarks to the Author):

Recommendation: Accepted

The following paper entitled "Record EMI shielding behavior of thin graphene/PMMA nanolaminates in the THz range" highlights graphene/PMMA composites synthesized by lift-off/float-on deposition process to produce multilayered laminates. In revision, authors have made substantial changes in the revised manuscript, and all the questions asked by the reviewer 1 have been explained comprehensively with good results. Therefore, we suggest the article for its publication in prestigious journal of Nature Communications.

Reply to Reviewers' Comments

We would like to thank Reviewer #3 for his/her final positive assessment of the revised manuscript and Reviewer #2 that gives us the opportunity to clarify further the mechanism of EMI shielding provided by the PMMA/graphene nanolaminates.

In the following the additional points raised by Reviewer #2 are fully addressed.

Comment 1

The graphene/polymer nanolaminates with an excellent shielding effectiveness up to 60 dB at a small thickness of 33 μm , which is a very exciting. Hence, the transmission process and loss mechanism of electromagnetic waves inside should be clearly revealed to help with the application in the high-frequency field.

Authors' reply

Indeed, the mechanisms governing the transmission and absorption in the graphene/polymer nanolaminates are very well described by the standard transmission line model of shielding [1]. Using this approach, the shielding effectiveness can be nicely predicted using the decomposition terms presented in eqs. S5 and S6. This is clearly seen by the very good agreement between the experimental data and the theoretical expectation shown in Fig. 1 of our previous reply to Reviewer #1 (reported below). During the different stages of the reviewing process, we had also the opportunity to perform reflection measurements, which confirm the Schelkunoff decomposition scenario and will be the subject of a forthcoming paper.

Figure 1. Comparison of experimental data of shielding effectiveness (\square) and the theoretical values (dotted line) for the 33 μm -thick laminate. SE_R , SE_A and SE_M (blue, red, and green continuous lines respectively) were calculated according to eqs. S5 and S6.

Comment 2

Why is it possible for jumping electrons to enhance the micro-current in the graphene layered package when the thickness of the PMMA spacer layer is less than 5 nm? How do the authors determine this thickness?

Authors' reply

We thank the reviewer for his/her comment. In this regard, we would like to emphasize that our attempt to shed light on the conduction mechanism was indeed enlightened by the comments/suggestions of Reviewer #2 and was, eventually, positively assessed by the same Reviewer after the first revision. First of all, for the sake of clarity, it is important to recall here that we fabricated PMMA layers with thickness t_{PMMA} ranging from 750 nm to 33 nm,

compatible with graphene content V_g in the nanolaminates from 0.04 to 1 vol% (Table S1). As shown in Figure S3 (optical and AFM images), the polymer layers conformally reproduce the morphology of the rough Cu foil and this induces inhomogeneities in the thickness of the layer itself, which are likely more pronounced when t_{PMMA} is comparable with the roughness of the Cu foil (70 nm ca.). As already stated in the previous reply to reviewers and in the revised manuscript, at small volume fractions ($t_{\text{PMMA}} > 100$ nm), the nanolaminate reasonably behaves as an ideal alternant-layered structure, and the electrical conduction can be ascribed to migrating electrons in the continuous π bonds of the graphene layers that are separated by the continuous insulating PMMA spacers. At higher graphene content (i.e. for $t_{\text{PMMA}} < 100$ nm), the presence of local thinning of the polymer spacer could be responsible of further conductive paths. In fact, according to several works in the literature [2,3,4,5], the cut-off distance for electron jumping between two parallel graphene sheets insulated by polymers is typically < 5 nm. Therefore, at high V_g , it is likely that the thickness of the PMMA spacer in some locations is close to the cut-off distance and electron jumping may be triggered. At this stage, conduction mechanism can be thus ascribed to the combination of migrating and hopping electrons [6,7,8,9,10].

In order to clarify this aspect, in the revised manuscript we have amended the discussion on the conduction in the Gr/PMMA nanolaminates and included references related to the cut-off distance, and the text now reads (changes are reported in red):

*“However, we observe that the linearity of in-plane conductivity with volume fraction is lost for higher graphene content (i.e. for $t_{\text{PMMA}} < 100$ nm), which may suggest the occurrence of further mechanisms contributing to charge transport. Actually, the aforementioned inhomogeneity of the polymeric layer thickness due to the use of a rough sacrificial substrate probably causes interconnection or bridging between adjacent layers. At this stage, therefore, it is possible that **the local thickness of the PMMA spacer is comparable with the cut-off distance for electron jumping between two parallel graphene sheets insulated by polymers ($< 5\text{nm}^{46-48}$) and therefore hopping electrons are triggered thus enhancing the micro-current in the layered package of graphene. Under these conditions therefore the conduction mechanism could be ascribed to the combination of migrating and hopping electrons**^{38,40,41,49}”*

Comment 3

Author think that, for the nanolaminates with a fixed graphene volume fraction (0.33 vol.%), losses dominate the shielding mechanism. However, it is generally believed that as the thickness increases, the conductivity will increase significantly. The high-efficiency EMI shielding should depend on reflection of electromagnetic wave. Therefore, I am curious about the authors' viewpoint. Is there any evidence for this?

Authors' reply

We thank the reviewer for his/her comment on this subtle point that actually needs to be better clarified. According to eqs. S5 and S6, both the absorption and reflection terms (SE_A and SE_R respectively) in the shielding mechanism have the same dependence on conductivity (square root), therefore by increasing σ they contribute the same way to the high-efficiency EMI shielding. Moreover, under the good conductor approximation, they contribute similarly - albeit with a different frequency dependence - to the overall shielding, as it can be gathered for example from Fig. S7 in supporting material. However, since the reflection term SE_R in the shielding is a surface phenomenon, the absorption term SE_A only (linearly) depends on thickness (eq. S6). Therefore, keeping the graphene volume fraction constant at 0.33 vol.%, from the Schelkunoff decomposition it is clear that, increasing the nanolaminate thickness from 5 to 33 μm , the relative contribution of losses undergoes a (more than) fivefold increase with respect to the reflection term. Experimentally, this is confirmed from the SE values being linearly dependent on thickness already at 1 THz (inset of Fig. 4c, main text). Once again, this can be also observed in Fig. 1 of our previous reply to Reviewer #1 (reported above), where the absorption term SE_A in the 0.33 vol.%, 33 μm thickness, sample dominates the shielding mechanism and governs its frequency dependence.

Comment 4-5

The authors give a detailed response to Reviewer #1's comments and corresponding modifications, and put forward to different viewpoint on the issue of AC conductivity (Comment d.). I think that the reviewer and the authors have a deviation in the understanding of the material object, which leads to a difference of opinion on the AC conductivity.

Drude's model is originated from the theory of free electron gas and is applicable to conductors such as graphene or graphene nanosheets. However, the Gr/PMMA reported in

this article is composite material, and its conduction mechanism follows the jumping electron model. Therefore, the change rule of the AC conductivity of these two is not necessarily same.

Authors' reply

We thank the Reviewer for his/her consideration. This has been dealt in our previous reply(ies) to Reviewer #1 which we are happy to reiterate here for the sake of completeness. Drude's model was invoked by Reviewer #1 (in his/her comment d) to point out to the fact that the a.c. electrical conductivity of graphene should theoretically decrease in the THz regime. We would like to stress that our disagreement was not based on a mere "academic" dispute but based on our experimental observations. Moreover, many results are available in literature (some of them [11-13] have been also included in the reference list of supporting material) confirming that graphene - in the THz region of interest - shows an a.c. electrical conductivity that is still Drude-like and almost flat in frequency.

We agree with the Reviewer (comment 5) that a free electron model cannot be rigidly applied to our Gr/PMMA composite material. Nevertheless, the linearity of in-plane conductivity with volume fraction still holds but at the highest graphene content. Therefore, in principle, a Drude scenario can work for our nanolaminates at small/intermediate graphene contents, in the stage where conduction mechanism is due to free electrons only. At higher volume fractions, conduction mechanism is due to combination of migrating and hopping electrons, therefore we agree with the Reviewer that the Drude's model might not be the best way to describe fully our system.

In order to take into account the Reviewer comments, the following text in the SI has been modified (changes are reported in red):

*"It is worth to highlight that in all measurements presented here the frequency behaviour of the shielding effectiveness is consistent with a simple Drude scenario where the graphene electrical conductivity still shows a flat response in the low THz region, consistently with many literature reports [16-18]. Of course, it is possible that a frequency drop in the **in-plane** electrical conductivity and therefore in the SE dependence occurs at higher frequencies, in agreement with the predictions given in [6]. **Moreover, as discussed in the main text, at the highest volume fraction there might be an additional contribution to the shielding given by the presence of hopping electrons (through-plane conduction) when t_{PMMA} is less than 100 nm.**"*

Comment 6

I'm curious about the shielding performance of this graphene/polymer nanolaminate in the GHz region. Authors may choose to answer this question.

Authors' reply

We agree with the reviewer on the importance of the shielding performance our nanolaminates may present in the GHz region, because of the variety of possible applications. Indeed, for the best performing samples we are confident that the Schelkunoff decomposition keeps its validity. Then, by assuming a constant graphene electrical conductivity, at 10 GHz we expect a tenfold decrease in SE_A and a tenfold increase in SE_R with respect to the results at 1 THz, according to their respective frequency dependence, to give an overall SE value of more than 30 dB. In such a case, the shielding mechanism is totally dominated by the reflection losses.

¹ S. A. Schelkunoff (1934). "The impedance concept and its application to problems of reflection, refraction, shielding and power absorption, Bell. Syst. Tech. J. 13, 532.

² Oskouyi A. B., Sundararaj U., Mertiny P. (2014), Tunneling Conductivity and Piezoresistivity of Composites Containing Randomly Dispersed Conductive Nano-Platelets, Materials, 7(4), 2501-2521.

³ Manta A., Gresil M., Soutis C. (2017). Predictive Model of Graphene Based Polymer Nanocomposites: Electrical Performance, Applied Composite Materials 24, 281–300.

⁴ Hicks J., Behnam A., Ural A. (2009). A computational study of tunnerring percolation electrical transport in graphene-based nanocomposites. Appl. Phys. Lett. 95, 213103:1-3.

-
- ⁵ Li, J., Kim, J.-K (2007). Percolation Threshold of Conducting Polymer Composites Containing 3D Randomly Distributed Graphite Nanoplatelets. *Compos. Sci. Technol.* 67, 2114–2120.
- ⁶ Wen, B., Cao, M. S., Hou, Z. L., Song, W. L., et al. (2013). Temperature dependent microwave attenuation behavior for carbon-nanotube/silica composites. *Carbon*, 65, 124-139.
- ⁷ Cao, M. S., Song, W. L., Hou, Z. L., Wen, B., & Yuan, J. (2010). The effects of temperature and frequency on the dielectric properties, electromagnetic interference shielding and microwave-absorption of short carbon fiber/silica composites. *Carbon*, 48(3), 788-796.
- ⁸ Marsden, A. J., Papageorgiou, D. G., Vallés, C., Liscio, A., Palermo, V., et al. (2018). Electrical percolation in graphene–polymer composites. *2D Materials*, 5(3), 032003.
- ⁹ Yousefi, N., Gudarzi, M. M., Zheng, Q., Aboutalebi, S. H., Sharif, F., & Kim, J. K. (2012). Self-alignment and high electrical conductivity of ultralarge graphene oxide–polyurethane nanocomposites. *Journal of Materials Chemistry*, 22(25), 12709-12717.
- ¹⁰ Yousefi, N., Sun, X., Lin, X., Shen, X., Jia, J., Zhang, B., ... & Kim, J. K. (2014). Highly aligned graphene/polymer nanocomposites with excellent dielectric properties for high-performance electromagnetic interference shielding. *Advanced Materials*, 26(31), 5480-5487.
- ¹¹ Tomaino JL, et al. (2011). Terahertz imaging and spectroscopy of large- area single-layer graphene. *Optics Express* 19, 141.
- ¹² Maeng I, et al. (2012). Gate-Controlled Nonlinear Conductivity of Dirac Fermion in Graphene Field-Effect Transistors Measured by Terahertz Time- Domain Spectroscopy. *Nano Letters* 12, 551.
- ¹³ Whelan PR, et al. (2018). Conductivity mapping of graphene on polymeric films by terahertz time-domain spectroscopy. *Optics Express* 26, 17749.

REVIEWERS' COMMENTS

Reviewer #2 (Remarks to the Author):

I would like to recommend revised manuscript to publish in Nature communications.

Your manuscript has been checked for clarity and against journal policies and formatting style. The issues listed below must be addressed; failure to do so will cause delays in acceptance.

For further information, please see our formatting instructions.

Please highlight all changes in the manuscript text file, either using the track changes feature in Microsoft Word or coloured highlighting in LaTeX.

Please include your response to these requests in the space provided and return this checklist with your final submission.

EDITORIAL REQUESTS:	AUTHOR RESPONSE:
POLICIES AND CHECKLISTS	POLICIES AND CHECKLISTS
An updated editorial policy checklist must be completed and uploaded as a related manuscript file with the revised manuscript. All points on the policy checklist must be addressed; if needed, please revise your manuscript in response to these points. Please note that this form is a dynamic 'smart pdf' and must therefore be downloaded and completed in Adobe Reader, instead of opening it in a web browser. https://www.nature.com/authors/policies/Policy.pdf	
TITLE PAGE (page 2 of our formatting instructions)	TITLE PAGE (page 2 of our formatting instructions)
To adhere to journal style, I suggest the following revision to the title. If you would like to suggest an alternative title, please ensure that it does not exceed 15 words and does not contain punctuation or acronyms. Please also avoid expressions such as 'record', 'unprecedented', etc.	We changed the title with the suggested one.
Effective electromagnetic interference shielding behaviour of thin graphene/ polymer nanolaminates in the THz range	
We do not allow a graphical abstract. Please either remove it or incorporate it into a figure in the main manuscript.	We removed it.
MAIN TEXT (pages 1 to 3 of our formatting instructions)	MAIN TEXT (pages 1 to 3 of our formatting instructions)
To comply with our article templates, the text must be split into: - Introduction (<1000 words), which must include the background and rationale for the work. The final paragraph should be a brief summary of the major results and conclusions. The results of the current study should only be discussed in this final paragraph. The Introduction should contain no references to figures or tables. - Results, which must be split into subheaded sections, ensuring that the subheadings are no longer than 60 characters including spaces. Subheadings should contain no punctuation. - Discussion, without subheadings. - Methods, which must be split into subheaded sections, ensuring that the subheadings are no longer than 60 characters including spaces. There is no word limit for this section.	The text has been split accordingly.
LANGUAGE AND STYLE (page 6 of our formatting instructions)	LANGUAGE AND STYLE (page 6 of our formatting instructions)
Please remove phrases such as 'record', 'for the first time',	They have been removed or

'unprecedented', etc., as novelty is clear from the context. Please also remove exaggerated language such as 'extremely', 'outstanding', 'impressive', etc.	rephrased.
Please do not use italics, bold font, underlining or speech marks unless required for technical terms (in the abstract, main text and display items).	Ok.
When using a space/point group notation to label Raman peaks throughout the main text, figures and Supplementary Information, please use italics for irreducible representations, whereas subscripts (g, u, 1, 2, etc.) should be typeset in roman font.	We have adopted the suggested notation of the Raman peaks labels.
Please make sure that mathematical terms throughout your manuscript and Supplementary Information (including in figures, figure axes, and legends) conform strictly to the following guidelines. Equations must be supplied in editable format, and not as images. Scalar variables (e.g. x , V , χ) must be typeset in italic, whereas multi-letter variables and functions (e.g. log) must be formatted in roman. Vectors (such as the wavevector k or the magnetic field vector B) must be typeset in bold without italics.	We confirm that the mathematics conforms to the guidelines.
METHODS AND DATA (page 3 of our formatting instructions)	METHODS AND DATA (page 3 of our formatting instructions)
Please rename the Methods section as 'Methods'.	Done.
Sufficient details of the experiments must be provided in the Methods section such that they could be reproduced without reference to published papers. Use of the term 'as described previously' is not encouraged.	Ok.
All published manuscripts reporting original research in Nature Research journals must include a data availability statement, as a separate section before the References and under the heading 'Data Availability'. The data availability statement must make the conditions of access to the "minimum dataset" that are necessary to interpret, verify and extend the research in the article, transparent to readers. This minimum dataset may be provided through deposition in public community/discipline-specific repositories, custom proprietary repositories or general repositories like Figshare, Zenodo and Dryad. Providing large datasets in supplementary information is strongly discouraged and the preferred approach is to make data available in repositories. Scientific Data, a Nature Research journal, maintains a list of approved and recommended data repositories to support researchers seeking suitable repositories for their data (https://www.nature.com/sdata/policies/repositories). Please refer to our authorship policy for information about authors' responsibilities for preserving and making available data, code and materials upon publication. Authors are responsible for obtaining all necessary permissions and ensuring compliance with local regulatory requirements for data sharing. The Data Availability Statement should also reference any source data published alongside the paper. If DOIs are provided, we also strongly encourage including these in the Reference list (authors, title, publisher (repository name),	We have amended the DAS.

identifier, year). For clinical datasets or third party data, please ensure that the statement adheres to our policy (https://www.nature.com/nature-research/editorial-policies/reporting-standards#availability-of-data)	
Please use the following template to provide all the information stated above: The XX data generated in this study have been deposited in the YY database under accession code ZZ [add hyperlink here]. The XX data are available under restricted access for {insert reason}, access can be obtained by {explain how}. The raw XX data are protected and are not available due to data privacy laws. The processed XX data are available at YY. The XX data generated in this study are provided in the Supplementary Information/Source Data file. The XX data used in this study are available in the YY database under accession code ZZ [Add hyperlink here].	
Nature Research policies (https://go.nature.com/data-availability-AIP) strongly encourage deposition of research data in public repositories. In some cases this is mandatory, and you may have been previously advised if that was the case. If you need help depositing and curating your research data you should consider:  - Contacting Springer Nature’s Research Data Helpdesk (https://go.nature.com/helpdesk-AIP) for advice - Finding a suitable data repository (https://go.nature.com/RD-policies-AIP) for your data Please provide a unique identifier for the data (for example a DOI or a permanent URL) in the data availability statement, if possible. If the repository does not provide identifiers, we encourage authors to supply the search terms that will return the data. For data that have been obtained from publicly available sources, please provide a URL and the specific data product name in the data availability statement. Data with a DOI should be included in the reference list and cited where relevant. Alternatively, include the data in the Supplementary Information. For datasets for which mandatory deposition is not required and the data can only be shared on request, please explain why in your Data Availability Statement and in your response here. Please refer to our data policies here: http://www.nature.com/authors/policies/availability.html	
DISPLAY ITEMS (pages 4 and 5 of our formatting instructions)	DISPLAY ITEMS (pages 4 and 5 of our formatting

	instructions)
The use or adaptation of previously published images is strongly discouraged. If this is unavoidable, please request the necessary rights documentation to re-use such material from the relevant copyright holders and return this to us when you submit your revised manuscript. Please check whether your manuscript or Supplementary Information contain third-party images, such as figures from the literature, stock photos, clip art or commercial satellite and map data.	We do not have used any previously published images.
Please include a title describing Figure 1. Figure titles should be brief, and ideally no longer than about one line, with minimal symbols. Titles should not contain punctuation and should not cite figure panels.	Done.
Any abbreviations, symbols or colours present in your figures must be defined in the associated legends.	We have defined the abbreviations in the legends.
In each Figure and Supplementary Figure where error bars are used, they must be defined.	Ok.
Please make sure that the terms 'atomic units (a. u.)' or 'arbitrary units (arb. units)' are appropriately used.	We have corrected it in Figure 2.
In the caption of Fig. 2 and in the Main Text please define FWHM and Pos(G).	Done.
In the caption of Fig. 3b please define SD.	Done.
SUPPLEMENTARY INFORMATION (page 5 of our formatting instructions)	SUPPLEMENTARY INFORMATION (page 5 of our formatting instructions)
We do not edit Supplementary Information files; they will be uploaded with the published article as they are submitted with the final version of your manuscript. Any tracked changes should be removed from the file and the file should be provided as a PDF file. Supplementary Figures do not need to be provided separately.	
The Supplementary Information should be organised using the following subheaded sections: - Supplementary Figures, labelled and referred to as Supplementary Figure 1, not Fig S1, throughout both the Supplementary Information and the main text - Supplementary Tables, labelled and referred to as Supplementary Table 1, not Table S1, throughout both the Supplementary Information and the main text - Supplementary Notes, numbered Supplementary Note 1, Supplementary Note 2, etc. - Supplementary Discussion - Supplementary Methods - Supplementary References, which should be self-contained.	We have structure the SI accordingly.
Supplementary References should appear at the end of the Supplementary Information file, and must be self-contained and numbered from 1. References mentioned in both the main text and the Supplementary Information should be part of both reference lists so that the Supplementary Information does not refer to the reference list in the main paper and vice versa.	

Please ensure all Supplementary References are formatted in the Nature Style. Please ensure that the format of all references follows the sequence: author list, title of paper, name of journal, volume number, initial-final page numbers (year).	We have formatted the references accordingly
All Supplementary figures should have a title briefly describing the whole figure.	Ok.
Please supply legends for each Supplementary Movie/Audio/Data file in your response here (not in the Supplementary Information file). Please label each files as Supplementary Movie/Audio/Data 1, etc.	Supplementary Movie: explainer video showing the iterative 'lift-off/ float-on' process combined with wet depositions adopted to produce the cm-scale Gr/PMMA nanolaminates.
PUBLICATION	PUBLICATION
Your paper will be accompanied by a two-sentence Editor's summary, of between 250-300 characters including spaces, when it is published online. I have drafted the summary below. If you would like to make changes to this, please provide me with a suitably edited version.	We are ok with the proposed summary.
The properties of graphene/polymer composites are usually limited by the use of discontinuous graphene flakes. Here, the authors report a fabrication method to realize continuous cm-scale graphene/polymer nanolaminates with enhanced electromagnetic interference shielding effectiveness, conductivity and mechanical properties.	
As part of our efforts to communicate our content to a wider audience, we endeavour to highlight papers published in Nature Communications on the journal's Twitter account (https://twitter.com/NatureComms). If you would like us to mention authors, institutions or lab groups in these tweets, please provide the relevant twitter handles.	The twitter handles is @costas_galotis